# Cross-Subject Modeling for Widefield Calcium Imaging via Atlas-Aligned Spatiotemporal Tokenization

**Mohammad Hosseini** [1]  **Eray Erturk** [1]  **Saba Hashemi** [2]  **Maryam M. Shanechi** [1 2 3]

## Abstract

Large-scale, multi-subject widefield calcium imaging provides unprecedented access to brain-wide cortical dynamics. However, the high dimensionality, complex spatiotemporal structure, and substantial task-irrelevant activity in widefield recordings have largely restricted modeling efforts to single-session analyses, limiting scalability and generalization. While multi-subject pretrained models have been explored for some neural modalities, multi-subject models for widefield calcium imaging have not yet been demonstrated; further, subject-invariant zero-shot behavior decoding remains elusive for multi-subject models across neural modalities more broadly. As a first step toward foundation modeling of widefield data, we introduce WiCAT, a multi-subject model that leverages self-supervised pretraining to both outperform single-session models and enable zero-shot behavior decoding on unseen subjects. WiCAT introduces an atlas-grounded tokenization scheme without session-specific components and learns globally shared spatiotemporal representations. Across multiple widefield datasets, the pretrained model supports lightweight downstream decoding, transfers across subjects, tasks, and datasets, and outperforms baseline models. Notably, the model also achieves robust zero-shot continuous behavior decoding and left-out brain region reconstruction on unseen subjects. Code: https://github.com/ShanechiLab/WiCAT/

[1]Ming Hsieh Department of Electrical and Computer Engineering, USC Viterbi School of Engineering, University of Southern California, Los Angeles, CA, USA [2]Thomas Lord Department of Computer Science, University of Southern California, Los Angeles, CA, USA [3]Alfred E. Mann Department of Biomedical Engineering, University of Southern California, Los Angeles, CA, USA. Correspondence to: Maryam M. Shanechi <shanechi@usc.edu>.

*Proceedings of the 43rd International Conference on Machine Learning*, Seoul, South Korea. PMLR 306, 2026. Copyright 2026 by the author(s).

## 1. Introduction

Widefield calcium imaging is an optical neural imaging modality that enables the recording of neural activity across large portions of the cortex, and in some settings near whole-cortex coverage, in behaving animals (Cardin et al., 2020; Saxena et al., 2020). By measuring fluorescence signals reflecting intracellular calcium dynamics, widefield imaging provides access to mesoscale neural activity with relatively high temporal resolution compared to imaging techniques that infer neural activity indirectly via vascular responses, while offering substantially broader spatial coverage than electrophysiological recordings (Nietz et al., 2022). Recent experimental efforts have produced unprecedented large-scale widefield datasets spanning multiple animals, recording sessions, behavioral tasks, and laboratories (Musall et al., 2019; Couto et al., 2021; Raut et al., 2025; Findling et al., 2025; Kondo et al., 2025). Thus, widefield imaging has become an important tool for studying distributed population dynamics, large-scale functional organization, and their relationship to complex behavior (Musall et al., 2019; Cardin et al., 2020; Ren & Komiyama, 2021; Nietz et al., 2022). Despite these advantages, widefield calcium imaging data present significant challenges for learning generalizable representations. Recordings are high-dimensional image time series with rich spatiotemporal structure, containing both task-relevant signals and substantial task-irrelevant or spontaneous activity (Musall et al., 2019; de Vries et al., 2020; Nietz et al., 2022; MacDowell et al., 2024; Hosseini & Shanechi, 2025). Given the above difficulties, existing approaches to modeling widefield imaging data are developed and applied independently for individual sessions or subjects, even when the datasets contain recordings from many animals (Saxena et al., 2020; Benisty et al., 2024; Karniol-Tambour et al., 2024; Hosseini & Shanechi, 2025). While effective within a single recording context, such approaches limit scalability and cross-subject generalization, hindering the reuse of learned representations across animals, sessions, and experimental conditions (Dinsdale et al., 2022).

One approach to addressing these challenges is to develop cross-subject modeling approaches for widefield imaging datasets that achieve shared representations and generalization across subjects and datasets. Indeed, while neural

dynamics and functional interactions between cortical regions evolve over time and with behavioral context, we note that cortical anatomy and large-scale spatial organization are highly consistent across subjects (Wang et al., 2020; Musall et al., 2023; Shahsavarani et al., 2023; Asadi et al., 2025; Kondo et al., 2025). This suggests that shared representations across animals are possible, but difficult to learn directly from raw widefield data. While recent multi-subject pretrained modeling approaches for other neural modalities such as spiking and local field potential (LFP) activity have demonstrated the benefits of learning shared representations across subjects and sessions (Ye et al., 2023; Azabou et al., 2023; Zhang et al., 2025; Erturk et al., 2025; Oganesian et al., 2025), such methods for widefield calcium imaging have not yet been explored. Moreover, these prior models typically rely on subject- or session-specific parameters that must be learned or retrained when transferring to new recordings, limiting true zero-shot generalization in behavior decoding for new subjects. Taken together, the question of whether learning shared representations and zero-shot transfer across subjects are viable for widefield neural imaging remains open, yet is critical to address as a first step toward foundation-style modeling for this modality.

**Contributions.** Here, we introduce WiCAT (Widefield Calcium imaging Atlas-aligned Tokenizer model), a multi-subject model trained on large-scale widefield calcium imaging data spanning neural recordings across 38 subjects, 378 sessions, and two distinct behavioral tasks. We leverage these diverse datasets to design an anatomically grounded tokenization scheme for widefield neural imaging together with a self-supervised pretraining approach that enables learning spatiotemporal representations that generalize across subjects, sessions, and behavioral tasks. Doing so, we establish a multi-subject model designed for widefield calcium imaging, in contrast to prior widefield methods that were restricted to single-subject modeling. We further demonstrate zero-shot continuous behavior decoding in unseen subjects, a capability that has remained elusive for multi-subject pretrained models across modalities.

Our main contributions are: **1)** We develop a multi-subject widefield imaging model that maps neural recordings from different subjects into a shared representation space. **2)** To do so, we propose an atlas-based tokenization and global spatial embeddings that enable cross-subject modeling without session-specific components, with learned embeddings that capture subject-invariant anatomical organization. **3)** We design a challenging self-supervised pretraining framework based on masked autoencoding (MAE) that learns spatiotemporal representations that reconstruct neural activity without access to subject or session identifiers. **4)** We demonstrate that these representations transfer effectively across subjects, tasks, and datasets, achieving competitive

performance using frozen representations and lightweight decoders. **5)** We show that, on unseen subjects, these representations support zero-shot continuous behavior decoding and efficient few-shot adaptation with limited labeled data, as well as zero-shot neural reconstruction of left-out brain regions, indicating that the learned representations capture cross-region dependencies that generalize across subjects.

## 2. Related Work

**Modeling Widefield Neural Images.** Widefield calcium imaging has motivated a broad range of approaches for behavior decoding and representation learning, including linear and nonlinear dynamical models (Batty et al., 2019; Benisty et al., 2024; Karniol-Tambour et al., 2024) as well as variational autoencoders (Wang et al., 2024). Due to the high dimensionality of widefield recordings, these methods predominantly rely on preprocessing steps such as PCA or atlas-guided decompositions (e.g., LocaNMF (Saxena et al., 2020)) to reduce dimensionality prior to downstream modeling. More recent end-to-end neural-behavioral dynamical models based on convolutional architectures have also been proposed to directly decode behavior from widefield recordings (Hosseini & Shanechi, 2025). These prior approaches are typically trained within a single recording session and require models optimized per session. Thus, they do not naturally scale to multi-subject or multi-session modeling and cannot leverage large-scale datasets to learn transferable representations. Our method, in contrast, enables multi-subject, multi-session modeling by mapping diverse recordings to a unified space using an atlas-grounded tokenizer.

**Multi-Session Modeling in Other Neural Modalities.** Outside the domain of widefield imaging, recent work has explored cross-subject modeling for other invasive direct recordings of brain activity, such as spiking activity (Ye et al., 2023; Azabou et al., 2023; Zhang et al., 2024), LFP (Erturk et al., 2025), and intracranial EEG (Wang et al., 2023; Zhang et al., 2023; Yuan et al., 2024; Mentzelopoulos et al., 2024; Zheng et al., 2024; Chau et al., 2025; Oganesian et al., 2025). These approaches employ various strategies to aggregate data across sessions, including unsupervised pretraining (Ye et al., 2023; Zhang et al., 2023; Chau et al., 2025; Oganesian et al., 2025; Erturk et al., 2025), supervised multi-task learning (Azabou et al., 2023; 2025), or hybrid objectives combining behavior decoding and neural reconstruction (Sani et al., 2021; 2024; Abbaspourazad et al., 2024; Zhang et al., 2025). A central challenge in these settings is handling distribution shifts across recordings. To address this, existing solutions typically introduce subject- or session-specific parameters, such as learnable session tokens (Ye et al., 2023; Zhang et al., 2025) or subject-specific read-in and read-out layers (Pandarinath et al., 2018; Mentzelopoulos et al., 2024). Consequently, these models

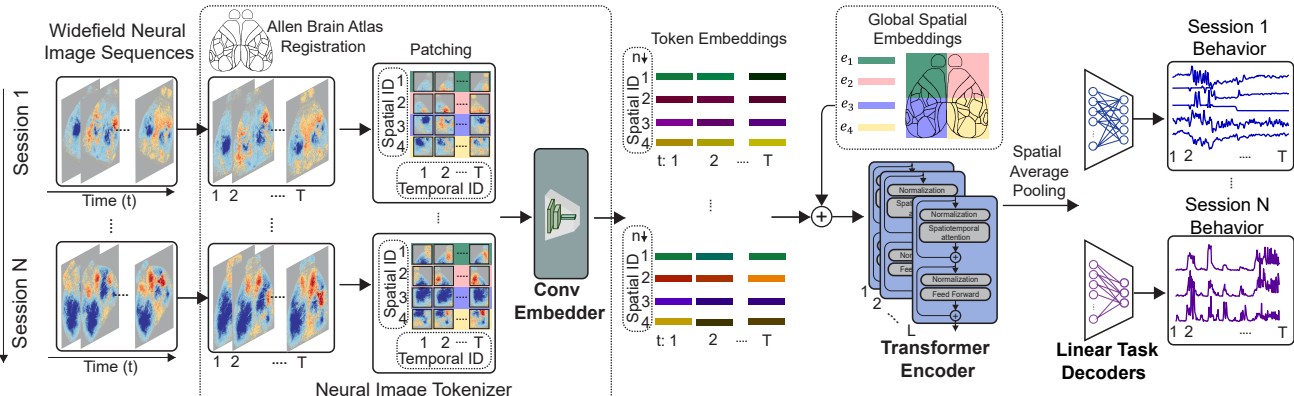

*Figure 1.* **Overview of WiCAT architecture.** Widefield calcium imaging frames are first registered to a common atlas (Allen Brain Atlas) and reshaped to a common spatial resolution of $128 \times 128$ to align cortical regions across subjects. **Tokenization and Embedding:** The aligned recordings are then partitioned into spatiotemporal patches. Each patch is projected into a $d$-dimensional latent space using a shared convolutional embedder. To enable cross-subject generalization, we learn a set of global spatial embeddings that are shared across all subjects and sessions and added to the token embeddings, providing a consistent representation of cortical location. For visualization clarity, the figure depicts a coarser $2 \times 2$ patch grid corresponding to an effective patch size of $64 \times 64$; however, in the actual model, a finer grid (e.g., 16 patches per frame with $32 \times 32$ patches) is used. **Transformer Backbone:** The resulting sequence of spatiotemporal tokens is processed by a Transformer encoder with joint spatial and temporal attention, using rotary positional embeddings (RoPE) to encode temporal information. **Downstream Decoding:** We apply spatial average pooling over latent embeddings from all spatial locations at each time point and feed the pooled representations to lightweight linear decoders for downstream tasks such as behavior decoding.

cannot robustly perform zero-shot generalization for behavior decoding in unseen subjects without finetuning. Finally, none of these models are developed for widefield imaging.

**Self-Supervised Pretraining.** Prior work has also explored self-supervised pretraining objectives for neural time series, including MAE (Ye et al., 2023; Zhang et al., 2023; Erturk et al., 2025), joint-embedding predictive architecture (JEPA) (Dong et al., 2024; Oganesian et al., 2025), combined with brain region spatial encoding for intracranial EEG in recent work (Oganesian et al., 2025), and contrastive learning (He et al., 2026). However, these approaches have not been explored for widefield calcium imaging, which consists of continuous spatiotemporal image sequences. Furthermore, it remains unclear what forms of self-supervised pretraining can enable zero-shot generalization to unseen subjects for widefield imaging without relying on session-specific parameters. Motivated by these gaps and to enable zero-shot generalization to unseen subjects, our work introduces an atlas-grounded tokenization scheme that aligns diverse subjects into a unified representation space and employs a challenging self-supervised pretraining objective to learn representations for widefield calcium videos that transfer across subjects, tasks, and datasets.

## 3. Methods

We introduce a multi-subject model for widefield calcium imaging designed to learn **shared representations across subjects and sessions** through unified multi-subject pre-

training (Figure 1). Our primary goal is to build a pretrained model that captures shared spatiotemporal representations in widefield data and supports strong downstream performance across recording sessions and animals, including **zero-shot generalization** in continuous behavior decoding and left-out brain region reconstruction on unseen subjects. To this end, we combine an atlas-grounded spatiotemporal tokenization scheme with a spatiotemporal Transformer backbone and a self-supervised masked reconstruction objective. Our tokenization approach combined with our pretraining objective together enable transfer to unseen subjects and sessions without introducing session or subject-specific parameters. After pretraining on multi-subject data, the learned representations are used in two evaluation settings: (i) training a linear decoder on a disjoint set of subjects with the backbone kept frozen, and (ii) zero-shot evaluation on entirely unseen subjects without any retraining.

### 3.1. Atlas-Grounded Tokenization of Widefield Imaging Data

**Neural Data Representation and Spatial Alignment.** We consider widefield calcium imaging recordings collected from $K$ subjects across $M$ sessions, potentially spanning multiple datasets with different recording setups, spatial resolutions, and cortical morphologies. A recording from dataset $r$ is initially represented as a spatiotemporal tensor $Y' \in \mathbb{R}^{T \times H_r \times W_r}$ where $T$ is the temporal length of the widefield recording segment, and $H_r$ and $W_r$ denote the dataset-specific spatial resolution.

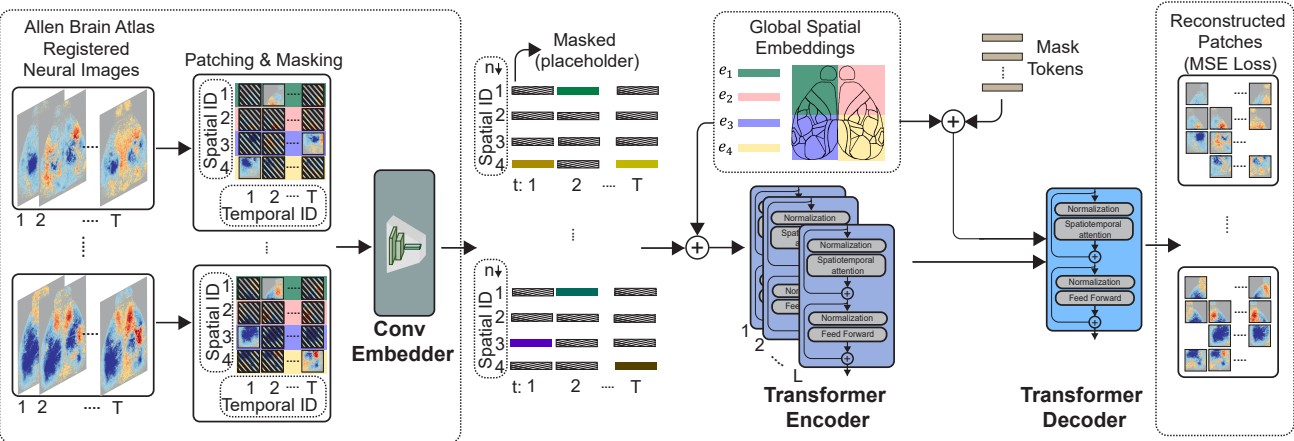

*Figure 2.* **Self-supervised pretraining via masked autoencoding.** Atlas-aligned widefield recordings are patchified into spatiotemporal tokens and a large fraction of tokens (90%) are masked across space and time. Unmasked tokens are embedded with a shared convolutional embedder and added to the global spatial embeddings before being processed by the Transformer encoder. Masked tokens are replaced by a learnable mask token, informed of the spatial location of the masked patch by adding the corresponding global spatial embeddings, and passed to a lightweight Transformer decoder, which reconstructs the masked patches using an MSE loss without using subject identifiers.

To enable parameter sharing across subjects and datasets, we spatially register all recordings to a common anatomical reference frame using the Allen Brain Atlas (Common Coordinate Framework, CCF) (Wang et al., 2020). After alignment to the Allen atlas, all datasets are resampled to a unified spatial resolution, resulting in tensors of consistent shape $T \times H \times W$ across subjects. This alignment step compensates for differences arising from distinct recording systems and is used solely to define a shared spatial coordinate system; importantly, it does not introduce any learnable subject-specific components.

**Spatiotemporal Tokenization.** Given an aligned recording, each frame is partitioned into non-overlapping spatial patches of size $P \times P$ (we use $P = 32$). Each patch at each timestep is treated as an individual spatiotemporal token, preserving the original temporal resolution without temporal downsampling. This yields $N = \frac{H \cdot W}{P^2}$ spatial tokens per timestep (the $H$ and $W$ dimensions are padded as necessary to ensure divisibility). We denote the resulting spatiotemporal patches as $\{y_{t,n}\}_{t=1,n=1}^{T,N}$, where $y_{t,n} \in \mathbb{R}^{P \times P}$.

**Global Spatial Embeddings.** Each patch $y_{t,n}$ is projected into a $d$-dimensional latent space using a shared learnable projection implemented as a spatial convolution applied independently to each patch. To encode anatomical identity in a subject-invariant manner, we introduce global spatial embeddings. Specifically, we associate each atlas-aligned patch index $n$ with a learnable embedding $e_n \in \mathbb{R}^d$ (Figure 1) and define the final token embedding as

$$z_{t,n} = f(y_{t,n}) + e_n,$$

where $f(\cdot)$ denotes the convolutional patch embedder. These atlas embeddings are shared across all time points, subjects, sessions, and datasets. By construction, this design avoids session or subject-specific spatial embeddings, enabling cross-subject transfer without finetuning.

**Temporal Encoding and Backbone.** The resulting spatiotemporal tokens $\{z_{t,n}\}$ are processed by a Transformer encoder that attends jointly across space and time. Specifically, tokens are flattened across the spatial and temporal dimensions to form a single sequence

$$Z = \{z_{t,n}\}_{t=1,n=1}^{T,N},$$

of length $N \times T$, where $N$ denotes the number of spatial patches and $T$ the temporal length of the widefield segment. Self-attention is applied over this unified sequence, enabling the model to capture interactions both across cortical regions and across time within a single attention mechanism.

Temporal order is incorporated using RoPE within the attention layers (Su et al., 2021), allowing the model to encode relative temporal information. Because the Transformer operates on sequences of tokens, this formulation naturally supports variable-length temporal sequences, enabling the model to process recording segments of different durations across tasks and datasets. Additional architecture details are provided in Appendix C.

### 3.2. Self-Supervised Pretraining via Masked Autoencoding

We pretrain the model using an MAE objective applied to the spatiotemporal neural token sequence (Figure 2). While recent latent-prediction frameworks (e.g., JEPA-style meth-

ods) have been explored in other domains, we find that a reconstructive objective is better suited for widefield calcium imaging (see Section 4.1 and Tables 1 and 6). Indeed, widefield recordings exhibit substantial trial-to-trial variability and spatially localized activity that are closely related to behavior (Musall et al., 2019; Benisty et al., 2024). Thus, preserving such fine-grained dynamics is key for downstream decoding. As such, we adopt a masked reconstruction pretraining objective.

To make the pretraining task more challenging and prevent overfitting to subject-specific noise or local correlations, we apply aggressive masking and randomly remove 90% of the spatiotemporal tokens across both space and time. Crucially, no session-, subject-, or dataset-specific identifiers are provided during pretraining. As a result, the model must rely solely on the sparse unmasked context and the shared global embeddings to reconstruct the missing regions, encouraging the model to learn global spatiotemporal dynamics rather than memorizing session-specific details.

To reconstruct the masked patches, we employ a single-layer Transformer decoder that takes as input the encoder embeddings together with learnable mask tokens corresponding to the masked patches. The mask tokens are informed of spatial location through the global atlas embeddings and of temporal position through RoPE. Reconstruction is performed using a mean squared error (MSE) loss between the reconstructed and original masked patches. We use a single-layer decoder to ensure that the encoder bears most of the representational burden during pretraining (Figure 2).

### 3.3. Datasets and Experiments

**Datasets.** We evaluate our framework on two publicly available widefield calcium imaging datasets. The Musall dataset (Churchland et al., 2019; Musall et al., 2019) consists of recordings from 13 mice across 26 sessions performing a delayed visual decision-making task. The Kondo dataset (Kondo et al., 2025) contains recordings from 25 mice across 352 sessions during an auditory lever-pull task. Detailed dataset statistics, preprocessing steps, and behavior label types used for decoding are provided in Appendix A.

**Evaluation Details.** To rigorously evaluate cross-subject generalization of the learned representations, we partition subjects into three disjoint sets: *pretraining*, *finetune*, and *zero-shot* (Figure 7 and Table 5). The model backbone is pretrained exclusively on the pretrain set. For behavior decoding, a linear decoder is trained on the *finetune set* with the backbone frozen (unless otherwise noted, e.g., in the case of full finetuning). Both the pretrained backbone and the decoder are then kept frozen and evaluated zero-shot on unseen subjects from the *zero-shot set*, without any gradient updates. Behavior decoding performance is quantified

using the coefficient of determination ($R^2$) with respect to ground-truth behavior. This three-split design ensures that performance reflects true generalization rather than overfitting to subject-specific patterns. We additionally evaluate few-shot adaptation and zero-shot neural reconstruction by reporting MSE on left-out brain regions in held-out subjects; detailed evaluation protocols are provided in Appendix B.

**Implementation Details** The model uses an 8-layer Transformer encoder with 8 attention heads and an embedding dimension of 512, a single-layer Transformer decoder, and a total of 36.0M trainable parameters. During self-supervised pretraining, 90% of the tokens are masked, such that the encoder processes only the remaining 10% of tokens. This results in total pretraining times of approximately 4 hours for the Kondo dataset and 2 hours for the Musall dataset using 4 NVIDIA RTX PRO 6000 Blackwell GPUs. Additional architectural, optimization, and training details are reported in Appendices C and D.

## 4. Results

### 4.1. Self-Supervised Pretraining Improves Cross-Subject Generalization

We first evaluate our model in a multi-subject setting by comparing against baselines (Appendix E), including those that use common widefield preprocessing pipelines such as PCA or LocaNMF, as well as SBIND—a recent single-session widefield imaging model (Hosseini & Shanechi, 2025). For completeness, we also develop a multi-session extension of SBIND to compare to. In this evaluation, the backbone is pretrained on a shared set of training sessions, and a decoder is trained on sessions from the *finetune set*. For all unsupervised methods, the pretrained backbone remains frozen and only the decoder is trained (e.g., WiCAT *Linear probing*). For WiCAT *Full finetuning*, we additionally report results with the backbone finetuned together with the decoder on the *finetune set*. Importantly, all finetuning is performed exclusively on the *finetune set*, and zero-shot subjects are never used during any finetuning—whether for WiCAT *Linear probing* or for WiCAT *Full finetuning*. We present the zero-shot results in the next section. Additional baseline and ablation details are provided in Appendix E.

As shown in Table 1, WiCAT achieves the strongest behavior decoding performance on the *finetune set* across both datasets, outperforming single-session baselines and multi-session methods. Using Wilcoxon signed-rank tests, WiCAT *Linear probing* significantly outperforms all baselines on both datasets ($p < 10^{-8}, n = 30$ for Musall and $p < 10^{-10}, n = 435$ for Kondo; largest p-values across baselines). Note that both multi-session variants of SBIND (SBIND-Sup and SBIND-Unsup) use the same Allen Brain Atlas alignment as WiCAT (Appendix E). Thus, compar-

*Table 1.* **Behavior decoding performance ($R^2$, Mean $\pm$ SEM).** Models are evaluated for both datasets either with training a linear decoder on the finetune set or by evaluating the frozen model and decoder on the zero-shot set without any finetuning. Within each column asterisk (*) denotes WiCAT is significantly better than all other baselines with p-value $< 1e - 6$ (Wilcoxon signed-rank test). Chance-level decoding baselines are reported in Appendix Table 10.

| Model | Musall Behavior Decoding $R^2 \uparrow$ | | Kondo Behavior Decoding $R^2 \uparrow$ | |
|---|---|---|---|---|
| | **Finetune Set** | **Zero-shot Set** | **Finetune Set** | **Zero-shot Set** |
| *Single-session baselines* | | | | |
| PCA + Linear Regression | $0.4141 \pm 0.0052$ | $0.0477 \pm 0.0307$ | $0.2762 \pm 0.0038$ | $0.0573 \pm 0.0060$ |
| MLP | $0.4600 \pm 0.0056$ | $0.0957 \pm 0.0124$ | $0.2946 \pm 0.0058$ | $0.1025 \pm 0.0015$ |
| SBIND-Sup (single-session) | $0.4294 \pm 0.0092$ | $0.1466 \pm 0.0096$ | $0.3061 \pm 0.0056$ | $0.0240 \pm 0.0018$ |
| WiCAT (single-session) | $0.4764 \pm 0.0060$ | $0.1249 \pm 0.0089$ | $0.3060 \pm 0.0042$ | $0.0399 \pm 0.0025$ |
| *Multi-session baselines* | | | | |
| LocaNMF + Linear Regression | $0.3592 \pm 0.0081$ | $0.1991 \pm 0.0303$ | $0.2260 \pm 0.0041$ | $0.0948 \pm 0.0074$ |
| SBIND-Unsup | $0.4632 \pm 0.0048$ | $0.1894 \pm 0.0182$ | $0.2738 \pm 0.0037$ | $0.1305 \pm 0.0042$ |
| SBIND-Sup | $0.4011 \pm 0.0054$ | $0.2444 \pm 0.0152$ | $0.2641 \pm 0.0043$ | $0.1547 \pm 0.0047$ |
| *Ablations* | | | | |
| WiCAT (JEPA pretraining) | $0.4980 \pm 0.0073$ | $0.2604 \pm 0.0136$ | $0.3072 \pm 0.0038$ | $0.1389 \pm 0.0051$ |
| WiCAT (Random initialization) | $0.4420 \pm 0.0082$ | $0.0936 \pm 0.0053$ | $0.2928 \pm 0.0046$ | $0.0985 \pm 0.0057$ |
| WiCAT (No pos enc) | $0.4848 \pm 0.0066$ | $0.1985 \pm 0.0083$ | $0.3343 \pm 0.0038$ | $0.1316 \pm 0.0051$ |
| WiCAT (Session-specific) | $0.4111 \pm 0.0040$ | $0.1025 \pm 0.0195$ | $0.2554 \pm 0.0036$ | $0.0301 \pm 0.0042$ |
| **WiCAT (Linear probing)** | $\mathbf{0.5124 \pm 0.0080^*}$ | $\mathbf{0.3274 \pm 0.0080^*}$ | $\mathbf{0.3396 \pm 0.0038^*}$ | $\mathbf{0.1840 \pm 0.0053^*}$ |
| **WiCAT (Full finetuning)** | $\mathbf{0.5783 \pm 0.0062}$ | $\mathbf{0.3675 \pm 0.0114}$ | $\mathbf{0.4245 \pm 0.0052}$ | $\mathbf{0.2398 \pm 0.0064}$ |

isons with multi-session SBIND indicate that spatial alignment alone is not sufficient for strong cross-subject generalization and that our atlas-grounded tokenization with globally shared spatial embeddings is critical for learning shared representations. We additionally adapt NDT2 (Ye et al., 2023) and CEBRA (Schneider et al., 2023) baselines, originally developed for other neural modalities, to the widefield calcium imaging modality (Appendix E); WiCAT also outperforms these baselines in additional comparisons reported in Appendix Table 7.

To isolate the role of the self-supervised objective, we also evaluate a JEPA self-supervised pretraining objective for our method (WiCAT *JEPA pretraining*) using the same tokenization and Transformer backbone. Despite extensive exploration of JEPA masking strategies used in prior work (Bardes et al., 2024; Dong et al., 2024), the learned representations using this approach had a significantly lower downstream behavior decoding performance than our pretraining framework (Tables 1, 6). We further find that, due to strong spatial and temporal correlations in widefield imaging data, masking 90% of spatiotemporal patches produces a sufficiently challenging self-supervised objective and achieves peak downstream decoding performance (Figure 10).

Finally, ablation results confirm that the core design choices of WiCAT are essential for learning globally shared spatiotemporal representations across subjects. Removing global spatial embeddings (WiCAT *No Pos Enc*) or introducing session-specific parameters (WiCAT *Session-specific*) significantly degrade performance on left-out subjects in the finetune set, showing that the pretrained representations

with these ablations do not generalize well to new subjects. In addition, variants trained without self-supervised training (WiCAT *Random initialization*) consistently underperform WiCAT, highlighting the importance of multi-subject pretraining for cross-subject generalization. Together, these results demonstrate that atlas-grounded tokenization and global spatial embeddings are key components that enable learning representations during self-supervised pretraining that generalize across subjects.

### 4.2. Zero-Shot Behavior Decoding on Unseen Subjects

To test zero-shot behavior decoding, after training the linear decoder on the *finetune set*, we freeze the entire model and evaluate it on sessions from the unseen subjects in the *zero-shot set*. For session-specific models that require retraining session-specific parameters, these parameters are initialized using embeddings learned for one of the sessions in the *finetune set*.

As shown in Table 1, WiCAT achieves substantially higher zero-shot decoding performance on the *zero-shot set* compared to all baseline models across both datasets ($p < 10^{-6}, n = 30$ for Musall and $p < 10^{-16}, n = 405$ for Kondo; Wilcoxon signed-rank test; largest p-values across multi-session baselines). In contrast, other baselines and ablated models with session-specific positional embeddings fail to generalize to new subjects. Note that similar to WiCAT *Linear probing*, WiCAT *full finetuning* also performs no gradient updates in the zero-shot set, thus again testing true zero-shot generalization. Together, these results indicate that the representations learned by WiCAT gener-

alize more effectively across subjects when trained with atlas-aligned tokenization, global spatial embeddings, and self-supervised pretraining.

**Additional downstream decoders.** While linear probing provides a controlled evaluation of the learned representations, it may underestimate task-relevant information in the frozen embeddings. We therefore evaluate lightweight linear dynamical decoders and nonlinear decoders, including Kalman filter variants and a one-layer MLP decoder, on top of WiCAT embeddings. For the Kalman decoder, we project frozen embeddings to a 16-dimensional latent state and predict behavior from this state (Appendix F.2). As shown in Table 2, Kalman smoothing improves zero-shot decoding from $R^2 = 0.3274$ to $0.4336$ on Musall and from $R^2 = 0.1840$ to $0.2869$ on Kondo, while the MLP decoder achieves comparable gains. These results indicate that WiCAT learns frozen representations with substantial predictive power in zero-shot cross-subject generalization, beyond what is revealed by simple linear probing and without requiring full backbone finetuning; applying the same decoder families to LocaNMF + CEBRA, NDT2, and SBIND yields smaller gains, suggesting that the improvement is not simply due to decoder capacity (Appendix Table 8).

*Table 2.* **Zero-shot decoder expressivity for WiCAT.** Zero-shot behavior decoding $R^2$ (mean $\pm$ SEM) on unseen subjects. All decoders except *Linear (Full FT)* use the frozen pretrained backbone. asterisk (*) indicates significantly better than others (Wilcoxon signed-rank test p $< 5e - 3$, n=sessions×seeds). The application of the same decoders to other baselines is reported in Appendix Table 8.

| Decoder | Musall Zero-shot $R^2 \uparrow$ | Kondo Zero-shot $R^2 \uparrow$ |
|---|---|---|
| Linear | $0.3274 \pm 0.0080$ | $0.1840 \pm 0.0053$ |
| Kalman Filter | $0.4313 \pm 0.0055$ | $0.2743 \pm 0.0053$ |
| Kalman Smoother | $\mathbf{0.4336 \pm 0.0056*}$ | $\mathbf{0.2869 \pm 0.0054*}$ |
| MLP | $0.4115 \pm 0.0229$ | $0.2867 \pm 0.0115$ |
| Linear (Full FT) | $0.3675 \pm 0.0114$ | $0.2398 \pm 0.0064$ |

**Few-shot adaptation.** We further evaluate few-shot adaptation on the *zero-shot set* by finetuning only the linear decoder with an increasing number of trials from each new subject. As shown in Figure 3, WiCAT requires fewer trials than the session-specific variant to adapt, reaching peak performance after approximately 16 trials (around 100 seconds of recording). In contrast, the session-specific variant starts from lower zero-shot performance and converges more slowly.

### 4.3. Cross-Dataset Transfer of the Pretrained Representations

We evaluate whether the representations learned by WiCAT transfer across datasets with different recording setups, spatial resolutions, and behavioral tasks. We pretrain the model on one dataset (source) and evaluate it on the other (target),

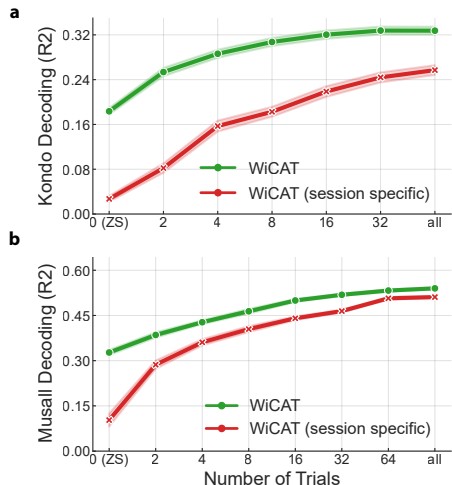

*Figure 3.* **Few-shot adaptation.** Behavior decoding results are shown for **(a)** Kondo and **(b)** Musall datasets for different numbers of trials used for adaptation. WiCAT reaches peak performance with fewer adaptation trials compared to WiCAT *Session-specific*. Each trial consists of 7.00 seconds (Kondo) or 6.83 seconds (Musall) of widefield image frames.

comparing against models pretrained directly on the target dataset.

*Table 3.* **Cross-dataset transfer for WiCAT.** Behavior Decoding $R^2$, Mean $\pm$ SEM across 5 finetuning seeds and sessions are reported for linear probing or fully finetuning the source model. Rows group within-dataset pretraining (*source = target*) and cross-dataset transfer (*source $\neq$ target*), while columns indicate the evaluation dataset.

| Linear Probing | | |
|---|---|---|
| Target / Source | **Musall** | **Kondo** |
| source = target | $0.5124 \pm 0.0080$ | $0.3396 \pm 0.0038$ |
| source $\neq$ target | $0.4550 \pm 0.0060$ | $0.2943 \pm 0.0041$ |

| Full Finetuning | | |
|---|---|---|
| Target / Source | **Musall** | **Kondo** |
| source = target | $0.5783 \pm 0.0062$ | $0.4245 \pm 0.0052$ |
| source $\neq$ target | $0.5672 \pm 0.0160$ | $0.4210 \pm 0.0105$ |

In the linear probing setting, the tokenizer and Transformer backbone are kept frozen, and only a linear decoder is trained on the target dataset. Despite this, as shown in Table 3, WiCAT achieves competitive cross-dataset performance under linear probing, comparable to baseline models trained on the target dataset (c.f. Table 1), indicating that the representations learned by WiCAT remain informative on a new dataset with a different behavioral task. We further evaluate full finetuning, where a model pretrained on the source dataset is finetuned on the target dataset. In this setting, models pretrained on a different source dataset perform comparably to models pretrained on the target dataset itself. This

suggests that pretraining on one widefield dataset provides a strong initialization that transfers effectively across datasets, further supporting the role of atlas-aligned tokenization and global spatial embeddings in unifying representations across subjects and recording conditions.

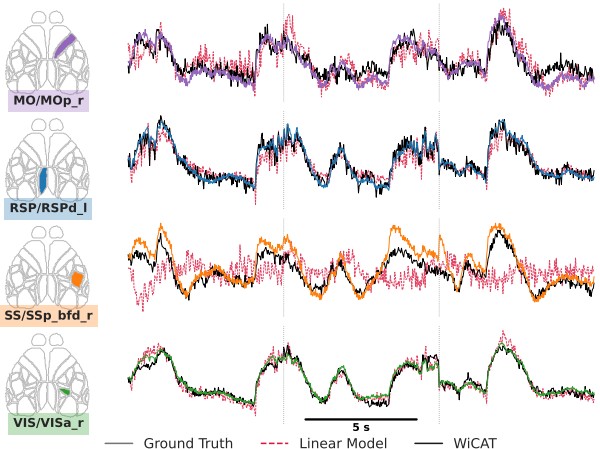

*Figure 4.* **Zero-shot neural reconstruction shows that our model effectively infers the activity of unseen functional areas.** Representative activity traces from four major cortical regions (MO, SS, VIS, RSP) for a held-out subject. The model's reconstructed neural dynamics in these regions closely track the ground truth (black) without any training on this subject. Despite using the same decoder for all regions, WiCAT's zero-shot reconstructions are more accurate than those of a linear model that is separately optimized for each brain region (dashed red). Additional per-pixel $R^2$ maps, spatial reconstruction snapshots, and temporal-horizon analyses are shown in Appendix Figures 11–13.

## 4.4. Zero-Shot Left-out Region Neural Reconstruction

Our pretraining framework naturally enables neural reconstruction. To evaluate this capability, we mask out entire brain regions and reconstruct their activity using only context from the remaining regions. Specifically, the Transformer encoder receives spatiotemporal tokens from all regions except the target region, while the decoder is provided with mask tokens informed by the global spatial embedding of the left-out region and tasked with reconstructing its activity (See Figure 2).

We compare WiCAT against a linear cross-region decoder baseline, trained separately for each target region to reconstruct it from all the remaining regions in the brain. This linear baseline is heavily parameterized and optimized per region, whereas WiCAT utilizes the same decoder for all regions, which is trained during self-supervised pretraining. We evaluate both approaches on held-out subjects in zero-shot and finetune sets that were never seen during training.

As shown in Table 4, WiCAT consistently achieves lower reconstruction error than the linear decoder across most brain regions on held-out subjects. Notably, the linear baseline exhibits substantial degradation for several regions, while

*Table 4.* **Zero-shot neural reconstruction** (pixel-wise MSE, Mean ± SEM) for left-out brain regions on unseen subjects. ROIs correspond to major cortical areas defined by the Allen CCF (See Figure 8). The **Average** reports reconstruction MSE across the 6 regions and 168 sessions. See Table 9 for neural reconstruction over a more detailed breakdown of brain regions. Chance-level neural reconstruction values are reported in Appendix Table 11.

| ROI | WiCAT ↓ | Linear Decoder ↓ |
|---|---|---|
| SS | $0.2914 \pm 0.0065$ | $0.7805 \pm 0.0309$ |
| VIS | $0.4963 \pm 0.0196$ | $0.5488 \pm 0.0222$ |
| MO | $0.2932 \pm 0.0070$ | $0.3080 \pm 0.0121$ |
| AUD | $0.0219 \pm 0.0008$ | $0.0209 \pm 0.0010$ |
| OB | $0.0298 \pm 0.0021$ | $0.0452 \pm 0.0026$ |
| RSP | $0.1990 \pm 0.0066$ | $0.2102 \pm 0.0075$ |
| **Average** | $\mathbf{0.2219 \pm 0.0064}$ | $0.3189 \pm 0.0109$ |

WiCAT maintains stable performance without any subject-specific finetuning. Qualitative reconstructions in Figure 4 further show that WiCAT accurately reconstructs neural activity in left-out cortical regions for unseen subjects, preserving region-specific temporal structure using only context from the remaining brain regions.

## 4.5. Learned Spatial Embeddings Capture Anatomical Organization

To examine whether WiCAT learns subject-invariant anatomical representations, we visualize patch embeddings after pretraining. For this analysis, we use a finer spatial tokenization with patch size 16, compared to the default patch size 32 used in the main model, to obtain more detailed cortical patches. We average embeddings over time and trials for each subject and patch, and apply PCA followed by t-SNE to visualize the resulting embeddings in two dimensions. In Figure 5, each point corresponds to one atlas-aligned patch from one subject. The resulting embedding space is strongly organized by cortical location rather than subject identity: patches from the same atlas-aligned location cluster together across subjects, anatomically neighboring patches remain nearby, and embedding correlations are higher for neighboring and left-right homologous brain regions (Appendix Figure 14). This indicates that WiCAT learns subject-invariant spatial embeddings that preserve shared cortical topology.

## 4.6. Scaling Laws

We investigated the scalability of our framework by analyzing behavior decoding performance as a function of pretraining data volume. As shown in Figure 6, performance improves for both *finetune* and *zero-shot sets* as the amount of pretraining data increases from 5% to 100%, with continued gains on the more challenging Kondo dataset. We also explored the effect of scaling model capacity. As detailed in Figure 9, increasing the Transformer backbone size

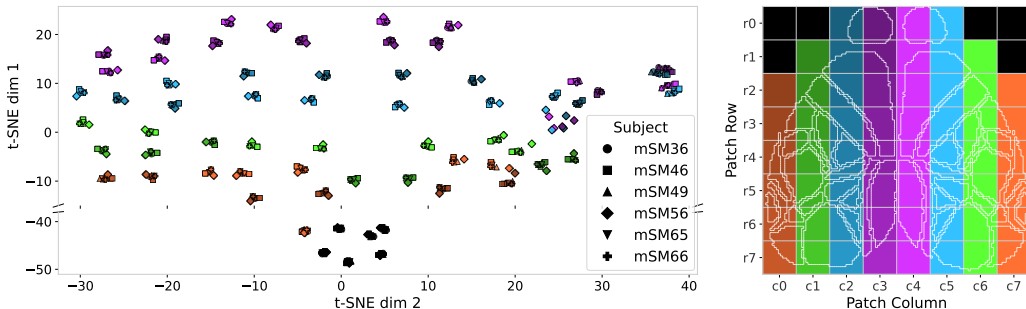

*Figure 5.* **t-SNE visualization of learned patch embeddings. (a)** Each point represents the mean embedding of one spatial patch for one subject, averaged over time and trials. Colors correspond to atlas-aligned patch location, and markers denote subjects. Patches from the same spatial location across subjects cluster together, while neighboring patches are embedded nearby, revealing subject-invariant anatomical topology. **(b)** Patch-coloring scheme used in (a).

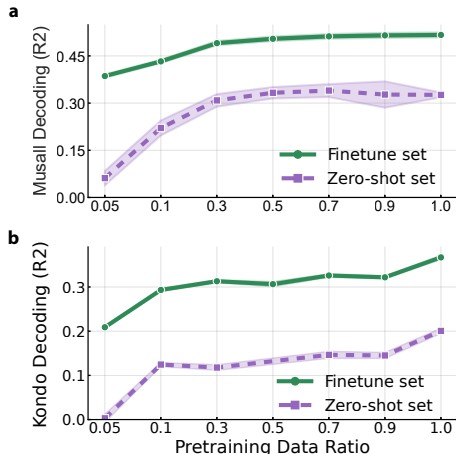

*Figure 6.* **Performance scales with pretraining data.** Behavior decoding performance ($R^2 \pm$ SEM, aggregated across 5 finetuning seeds and all sessions) for finetune and zero-shot sets on **(a)** Musall and **(b)** Kondo datasets as a function of pretraining data ratio.

similarly results in performance improvements. These results indicate that the proposed atlas-aligned tokenization and self-supervised pretraining framework can effectively leverage additional data and parameters to learn richer spatiotemporal representations of widefield neural activity.

## 5. Discussion

Here, we developed a cross-subject modeling framework for widefield calcium imaging that enables generalization across subjects, sessions, and datasets, in contrast to prior widefield methods that were restricted to single-session or single-subject modeling. By combining an atlas-grounded tokenization scheme and global spatial embeddings with a self-supervised masked autoencoding objective, our approach enables parameter sharing across animals without requiring session-specific components or retraining. Using this framework, we showed that representations learned from large-scale multi-subject data outperform several strong baselines, and further support zero-shot continuous behavior decod-

ing and left-out brain region reconstruction in held-out animals. Taken together, this work provides a first step toward foundation-style models of widefield imaging as larger and more diverse datasets become available.

Several directions for future improvement remain. Although we leverage some of the largest publicly available widefield datasets, the overall scale of publicly available widefield data remains smaller than in some other modalities and is largely restricted to head-fixed paradigms reflecting current hardware constraints; future work can increase the overall data scale to further improve performance and zero-shot generalization. Also, we use frozen representations to isolate the representational quality learned during pretraining, and show that a simple Kalman dynamical decoder improves temporal consistency and zero-shot prediction performance over linear probing. Future work can investigate richer adaptation strategies, including partial finetuning and cross-attention decoders over spatiotemporal tokens for downstream tasks. Finally, WiCAT relies on atlas alignment to provide a shared coordinate system across subjects. This atlas alignment facilitates cross-subject transfer for widefield imaging. Our perturbation analysis (Appendix Table 12) shows that WiCAT tolerates moderate affine perturbations, but larger misalignments degrade zero-shot transfer. Future work could improve robustness via alignment augmentations or registration-aware objectives during pretraining.

Beyond widefield calcium imaging, advances in simultaneous recording of optical imaging and other modalities (Lake et al., 2020; Vafaii et al., 2024) motivate multimodal neural modeling that integrates local neuronal spiking activity with cortex-wide population dynamics (Peters et al., 2021; MacDowell et al., 2025). Developing such models – analogous to multimodal learning in vision and language like CLIP (Radford et al., 2021) – could elucidate how neural activity is coordinated across spatial scales, as also shown in spike-LFP multimodal models and distillation approaches (Erturk et al., 2025; Erturk & Shanechi, 2025; Abbaspourazad et al., 2021). Finally, modeling the effect of external inputs is an important future direction (Vahidi et al., 2024; 2025).

## Acknowledgment

This work was supported, in part, by National Institutes of Health (NIH) grants R01MH123770, R61MH135407, and RF1DA056402, National Science Foundation (NSF) CR-CNS program award IIS-2113271, the Raynor Cerebellum Project (RCP), the One Mind Rising Star Award, and the Foundation for OCD Research (FFOR). Finally, we thank all members of NSEIP Lab for their helpful comments, discussions, and feedback throughout this work.

## Impact Statement

This paper presents work whose goal is to advance the fields of machine learning and neuroscience by developing a method for modeling large-scale neural imaging data across subjects. While such methods may positively inform future research directions, including neural decoding or brain-computer interfaces, this work focuses on methodological advances, and we do not anticipate specific societal or ethical impacts beyond those common to data-driven modeling approaches.

## Reproducibility Statement

To ensure the reproducibility of our work, we are sharing the model weights and code for WiCAT at `https://github.com/ShanechiLab/WiCAT/`. We also provide model hyperparameters and training details in Appendix C. The Widefield Imaging datasets (Churchland et al., 2019; Kondo et al., 2025) used in our main experiments are publicly available; details of the preprocessing applied to these datasets are provided in Appendix A, and our preprocessing framework for these datasets is also shared in our repository for those interested in reproducing the main experiments presented in this work.

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

# A. Dataset Details

We evaluate our model on two publicly available widefield calcium imaging datasets collected from behaving mice: the Musall dataset (Churchland et al., 2019; Musall et al., 2019) and the Kondo dataset (Kondo et al., 2025). Both datasets provide large-scale recordings of cortex-wide neural activity across multiple animals and sessions, together with continuous behavioral measurements.

## A.1. Musall Dataset

The Musall dataset consists of widefield calcium imaging recordings from head-fixed mice performing an auditory or visual decision-making task (Musall et al., 2019). Mice were trained to report the spatial location of 0.6 s-long sequences of auditory click sounds or visual LED stimuli by licking the corresponding spout. Trials were initiated by a handle grab, followed by stimulus presentation, a delay period, and a response period during which correct choices were rewarded with water.

Neural activity was recorded across the dorsal cortex using widefield calcium imaging at a sampling rate of 30 Hz. Raw images were originally recorded at a spatial resolution of $512 \times 512$ pixels. Behavioral data were obtained from behavior video recordings using two cameras capturing the animal's face and body. Continuous behavioral variables were extracted from predefined regions of interest, including jaw movement, nose position, whisking activity, and pupil diameter (Figure 17). These behavioral traces were used as continuous targets for decoding.

## A.2. Kondo Dataset

The Kondo dataset contains widefield calcium imaging recordings collected from mice performing an auditory cue-guided lever-pull task (Kondo et al., 2025). Trials were initiated by presentation of an auditory tone cue. A trial was considered successful if the mouse pulled a lever for longer than a specified duration within 1 s of cue onset, after which a water reward was delivered.

Widefield calcium images were recorded at a sampling rate of 30 Hz at a spatial resolution of $288 \times 288$ pixels. Behavioral data were obtained from behavior video recordings from three camera views capturing facial features, paw movements, and body posture. Continuous kinematic variables were extracted, and we use a total of 12 behavioral dimensions for decoding (Figure 19).

## A.3. Preprocessing

For both datasets, neural images were corrected for hemodynamic artifacts using dual-wavelength excitation recordings. Specifically, blue-excitation and violet-excitation images were acquired at each timestep. First, we temporally filtered both sets of images using a Butterworth high-pass filter (filtfilt with a cutoff frequency of 0.2 Hz, order 2). Following an established preprocessing step (Musall et al., 2019; Hosseini & Shanechi, 2025; Kondo et al., 2025; Sharafi et al., 2026), we orthogonalized the blue-excitation signal with respect to the violet-excitation signal using linear regression to remove contamination from hemodynamic signals.

We spatially aligned widefield images to the Allen Brain Atlas using per-session affine transformations, including translation, rotation, and scaling. After alignment, all recordings from both the Musall and Kondo datasets were resampled to a common spatial resolution of $128 \times 128$ pixels.

For the Musall dataset, recordings were segmented into fixed-length, trial-aligned temporal windows of $T = 205$ frames. In contrast, trials in the Kondo dataset exhibit variable durations and are less temporally structured, as animals do not consistently initiate or complete the lever-pull on every trial. To accommodate this variability, we segmented Kondo recordings into fixed 7-second windows ($T = 210$ frames at 30 Hz), spanning from 2 seconds before trial onset to 5 seconds after onset. This window captures both pre-trial and post-onset neural activity, even when behavioral execution is incomplete or absent.

## A.4. Dataset Splits

Each dataset was partitioned into disjoint subsets for self-supervised pretraining (pretrain set), decoder training (finetune set), and zero-shot evaluation (zero-shot set). For all sessions of the pretrain set, 80% of the data was used for self-supervised pretraining. For all sessions of the finetune set, 50% of the data were used for evaluation, while the remaining data were used

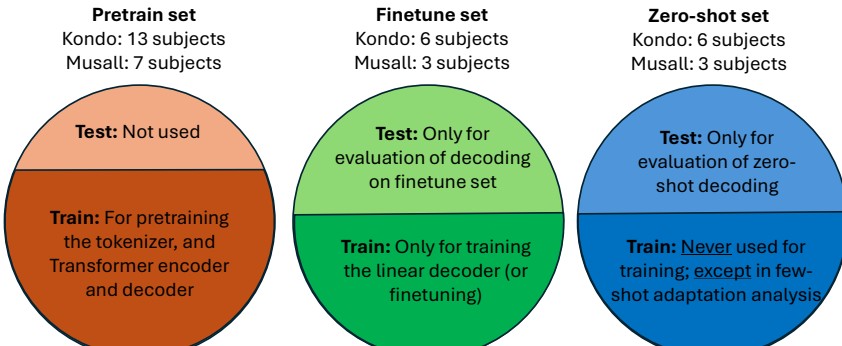

*Figure 7.* **Schematic illustration of the disjoint dataset splits and their roles in training and evaluation.** For each split, lighter colors indicate the test set, while darker colors indicate the training and validation data used for model optimization. The pretrain set is used exclusively for self-supervised pretraining of the tokenizer, Transformer encoder, and Transformer decoder. The finetune set is used only for training (or finetuning) the behavior decoder and to evaluate linear probing (or finetuning) performance. The zero-shot set is never used during training and is reserved exclusively for zero-shot evaluation, except in few-shot adaptation analyses.

for training and validation. For all sessions of the zero-shot set, 50% of the data were used for evaluation; the other 50% was only used for few-shot adaptation experiments to finetune the linear decoder (Figure 7). Dataset statistics, including the number of subjects and sessions in each set, are reported in Table 5.

*Table 5.* **Summary statistics for the widefield calcium imaging datasets.** We report the number of subjects and sessions used for each split, as well as the total number of segments and recording duration after preprocessing.

| Statistic | Musall dataset | Kondo dataset |
|---|---|---|
| Total subjects | 13 | 25 |
| Pretraining subjects | 7 | 13 |
| Finetuning subjects | 3 | 6 |
| Zero-shot subjects | 3 | 6 |
| Total sessions | 26 | 352 |
| Pretraining sessions | 14 | 184 |
| Finetuning sessions | 6 | 87 |
| Zero-shot sessions | 6 | 81 |
| Total segments | 11729 | 35738 |
| Total recording duration (hours) | 22.26 | 69.49 |

# B. Evaluation Details

In this section, we describe in detail the evaluation protocols used for each set of experiments reported in Section 4. All evaluations are performed using the dataset splits described above, namely the *pretrain set*, the *finetune set*, and the *zero-shot set* (Figure 7). These protocols are designed to isolate the effects of self-supervised pretraining, assess cross-subject generalization, and ensure fair comparison across baseline methods.

## B.1. Decoder Training on the finetune set

Results reported in the **finetune set** columns of Table 1 correspond to training a behavior decoder on the finetune set of each dataset. For all methods listed in Section 4.1 and Table 1, we report $R^2$ for behavior decoding, averaged across sessions in the finetune set and across 5 finetuning seeds (mean ± SEM).

For single-session models (Linear regression, MLP, SBIND and WiCAT *single-session*), a separate model is trained independently for each session in the *finetune set*. Each model is optimized using the training split of that session and evaluated on its corresponding test split. For multi-session SBIND and WiCAT *random initialization*, a single decoder is trained jointly using all sessions and subjects in the *finetune set*. For all ablations, and unsupervised SBIND, the model backbone is first pretrained on the pretrain set. The pretrained backbone is then frozen, and only a single linear decoder

is trained using the finetune set. This evaluation isolates the contribution of self-supervised pretraining and multi-session decoder training across baselines and ablations, enabling a fair and direct comparison of representation quality relative to WiCAT.

## B.2. Zero-Shot Evaluation on Held-Out Subjects

Results reported in the **zero-shot set** columns of Table 1 evaluate cross-subject generalization without any access to data from the zero-shot subjects during self-supervised pretraining or decoder training. In this setting, the same decoder trained on the finetune set is applied directly to the zero-shot set.

For single-session models, the decoder trained on each finetuning-session model is evaluated zero-shot on all zero-shot sessions. Behavior decoding $R^2$ is averaged across all finetuning-session decoders, zero-shot sessions, and random seeds. For multi-session models, the single decoder trained on the finetune set is evaluated directly on all sessions and subjects in the zero-shot set. This explicitly measures the ability of a model to generalize learned representations to entirely unseen subjects without any retraining of the backbone architecture and the linear decoder.

## B.3. Few-Shot Adaptation on Zero-Shot Subjects

In addition to zero-shot evaluation, we report few-shot adaptation results for WiCAT and its session-specific variant. In this setting, the decoder trained on the finetune set is further finetuned using a limited number of labeled trials from each session in the zero-shot set.

For each zero-shot subject, we finetune only the linear decoder using a small number of trials, while keeping the pretrained backbone frozen. For WiCAT *session-specific*, we train the session-specific spatial positional embeddings of the new sessions along with the finetuning of the linear decoder. Results are averaged across sessions in the zero-shot set and 5 few-shot seeds. This experiment evaluates how quickly each method adapts to new sessions and quantifies the gap between zero-shot and full-shot decoding performance.

## B.4. Cross-Dataset Transfer Evaluation

To assess cross-dataset generalization, we evaluate how representations learned on one dataset transfer to another dataset with different recording conditions and behavioral paradigms. For this experiment, the model backbone and tokenizer are pretrained exclusively on the pretrain set of a *source dataset*. The pretrained model is then transferred to a *target dataset*, where a single linear decoder is trained on the finetune set of the target dataset.

Results are reported by averaging behavior decoding $R^2$ across sessions in the target finetune set and across 5 finetuning seeds. This evaluation isolates how well representations learned on the source dataset, without any exposure to target data during pretraining, can support decoding using a lightweight linear decoder.

We additionally report results for **full finetuning**, where both the backbone and decoder are finetuned on the target dataset. This setting provides an estimate of the upper-bound performance achievable when initializing from a pretrained model learned on a different source dataset.

## B.5. Zero-Shot Neural Reconstruction

For neural reconstruction experiments, we evaluate whether WiCAT learns transferable spatiotemporal cortical dynamics that generalize across subjects. In this setting, the model is pretrained on the pretrain set and the decoder used during self-supervised pretraining is retained.

We then evaluate reconstruction performance on subjects never seen during decoder training, including all sessions from both the finetune set and the zero-shot set. For each subject, entire brain regions are masked and reconstructed using only activity from the remaining regions. This protocol evaluates the model's ability to capture cross-region neural dependencies in unseen subjects, without any subject-specific adaptation.

Together, these evaluation protocols provide a comprehensive assessment of representation quality, cross-subject generalization, cross-dataset transfer, and the ability to model large-scale spatiotemporal neural dynamics in widefield calcium imaging.

## C. Model Architecture and Self-Supervised Pretraining

**Tokenizer and Transformer Backbone.** As described in Section 3.1, widefield calcium imaging recordings are first aligned to the Allen Brain Atlas and partitioned into non-overlapping spatial patches of size $32 \times 32$ at each timestep, yielding 16 spatial patches per frame. Each patch is embedded using a shared convolutional patch embedder consisting of a single convolutional layer with 512 output channels and a kernel size matching the patch dimensions. A learnable global spatial embedding $e_n \in \mathbb{R}^{512}$, indexed by spatial patch location, is added to each patch embedding to encode anatomical location in a subject-invariant manner.

The resulting spatiotemporal tokens are flattened across space and time and processed by an 8-layer Transformer encoder with an embedding dimension of 512. The Transformer attends jointly across spatial and temporal dimensions, allowing each token to integrate information from all cortical regions and time points within the input segment. Relative temporal information is encoded using RoPE in all self-attention layers (Su et al., 2021). No session- or subject-specific parameters are introduced in either the tokenizer or the Transformer backbone.

**Masking Strategy and Self-Supervised Pretraining.** Self-supervised pretraining is performed using an MAE objective (He et al., 2021; Tong et al., 2022). During pretraining, 90% of spatiotemporal tokens are randomly masked across both space and time, such that the encoder processes only the remaining 10% of tokens. This masking ratio was selected based on downstream decoding performance on the validation split of the *finetune set* (Figure 10), which also reduces computational cost by limiting the number of tokens processed by the encoder.

Masked tokens are replaced with a learnable mask embedding and, together with the encoder outputs, passed to a lightweight single-layer Transformer decoder with embedding dimension $d = 512$. A linear projection head then maps the decoder outputs back to the input patch space ($32 \times 32$ pixels) to reconstruct the masked patches using an MSE loss. This asymmetric design places most representational capacity in the encoder while keeping the decoder lightweight.

**Downstream Decoder.** For all downstream evaluations, we use a simple linear decoder. The decoder operates on the encoder outputs, which retain a spatiotemporal structure of $N \times T$ tokens, where $N$ denotes the number of spatial patches and $T$ the temporal length of the segment. To obtain inputs suitable for behavior decoding, we apply average pooling across the spatial dimension (See Figure 1), yielding a $T \times d$ representation for each segment. No temporal pooling is applied, as downstream tasks involve decoding continuous behavioral signals.

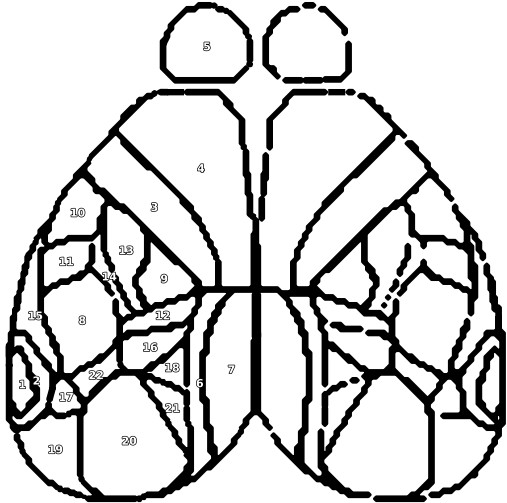

*Figure 8.* **Cortical areas based on the Allen CCF.** The map highlights anatomical regions with boundaries. Region correspondences (left hemisphere) are: **1**: Primary auditory area (AUDp); **2**: Secondary auditory area (AUDs); **3**: Primary motor area (MOp); **4**: Secondary motor area (MOs); **5**: Olfactory bulb (OB); **6**: Retrosplenial area, lateral agranular part (RSPagl); **7**: Retrosplenial area, dorsal part (RSPd); **8**: Primary somatosensory area, barrel field (SSp_bfd); **9**: Primary somatosensory area, lower limb (SSp_ll); **10**: Primary somatosensory area, mouth (SSp_m); **11**: Primary somatosensory area, nose (SSp_n); **12**: Primary somatosensory area, trunk (SSp_tr); **13**: Primary somatosensory area, upper limb (SSp_ul); **14**: Primary somatosensory area, unassigned (SSp_un); **15**: Supplemental somatosensory area (SSs); **16**: Anterior visual area (VISa); **17**: Anterolateral visual area (VISal); **18**: Anteromedial visual area (VISam); **19**: Lateral visual area (VISl); **20**: Primary visual area (VISp); **21**: Posteromedial visual area (VISpm); **22**: Rostrolateral visual area (VISrl).

## D. Training Details and Hyperparameters

**Pretraining.** All models were pretrained using distributed training on 4 NVIDIA RTX PRO 6000 Blackwell Server Edition GPUs with flash attention (Dao et al., 2022). We used an effective batch size of 128 and trained the model for up to 500 epochs with an early stopping patience of 50 epochs, selecting the best-performing checkpoint based on the reconstruction MSE on the validation split. The full model used in all experiments contains approximately 36.0M learnable parameters.

Across both datasets, pretraining was performed on approximately 72.1 million spatiotemporal tokens, corresponding to 3.24 million timesteps for the Kondo dataset and 1.26 million timesteps for the Musall dataset (51.9 million and 20.2 million tokens, respectively). Pretraining took $\sim 230$ minutes for the Kondo dataset and $\sim 110$ minutes for the Musall dataset.

We employed the AdamW optimizer (Loshchilov & Hutter, 2017) with a learning rate schedule consisting of a linear warmup phase over the first 50 epochs, starting at 0.3 of the maximum learning rate and increasing to a peak learning rate of $6.25 \times 10^{-4}$ (Goyal et al., 2017). This was followed by exponential decay with a decay factor of 0.995. Weight decay was linearly increased from 0.1 to 0.25 over the course of training.

**Downstream Decoding and Finetuning.** For downstream behavior decoding, we trained linear decoders on top of the frozen pretrained backbone for up to 300 epochs with an early stopping patience of 30 epochs, using the same optimizer and learning rate schedule as in pretraining. For experiments involving full backbone finetuning, we trained the entire model for up to 300 epochs, where the new parameters and the finetuning parameters were optimized with maximum learning rates of $6.25 \times 10^{-4}$ and $10^{-5}$, respectively.

In few-shot adaptation experiments on zero-shot subjects, we set the batch size to half the number of available trials, ensuring an equal number of optimization steps across few-shot finetuning experiments.

## E. Baselines and Ablations

We compare WiCAT against a diverse set of baselines spanning (i) end-to-end neural-behavioral dynamical models, (ii) self-supervised representation learning approaches, and (iii) commonly used preprocessing and decoding pipelines for widefield imaging data. We additionally include ablations of our model to isolate the contribution of each design choice.

**SBIND.** SBIND (Hosseini & Shanechi, 2025) is a state-of-the-art neural-behavioral dynamical model designed for widefield calcium imaging. SBIND is an end-to-end convolutional recurrent architecture that combines ConvRNNs with self-attention to model local and long-range spatiotemporal dependencies in neural images. The model is trained in two stages: a first stage that learns behaviorally relevant neural dynamics using supervised neural-behavioral prediction, followed by a second stage that models residual neural dynamics using an additional ConvRNN.

To further investigate whether the tokenization strategy in WiCAT is critical for cross-subject generalization, we also extend SBIND to a multi-subject, multi-session setting by aligning all recordings to the Allen Brain Atlas using the same spatial alignment procedure as WiCAT. We evaluate two variants: **SBIND-Sup** corresponds to the fully supervised version trained end-to-end on the finetune set, using the first stage of SBIND. **SBIND-Unsup** corresponds to the unsupervised variant obtained by disabling the first stage ($n_1 = 0$ in the original formulation), which disables supervision and serves as a self-supervised pretraining method. For SBIND-Unsup, the backbone is pretrained using the one-step-ahead neural prediction objective, and a separate downstream convolutional decoder is trained for behavior decoding without finetuning the backbone. Both variants use the original SBIND architecture and training protocol adapted to our datasets.

We use the best hyperparameters reported in the original SBIND (Hosseini & Shanechi, 2025) for all experiments. In particular, we use a latent dimensionality of $n_x = 32$ and a self-attention patch size of 8, together with the same convolutional architectures for the neural encoder, neural decoder, and behavioral decoder (3, 3, and 4 convolutional layers, respectively), using identical kernel sizes, channel dimensions, and other hyperparameters.

**PCA + Linear Regression.** We include a linear regression baseline reflecting a standard pipeline in widefield calcium imaging, which serves as a widely used reference for behavior decoding. For each session, neural images are first projected onto the top 500 principal components using principal component analysis (PCA), to mitigate noise and high dimensionality while retaining most of the variance in the images (Musall et al., 2019). A linear regression model is then trained on the resulting low-dimensional time series to predict behavior.

**LocaNMF-based baselines.** We include LocaNMF (Saxena et al., 2020), a standard atlas-guided decomposition pipeline for widefield calcium imaging. LocaNMF components are learned on the pretraining set and then applied to the finetune and zero-shot sets using the same atlas alignment. We evaluate two downstream decoders on the resulting low-dimensional activity: linear regression (LocaNMF + Linear Regression) and CEBRA (Schneider et al., 2023) followed by downstream decoding (LocaNMF + CEBRA). Both decoders are fit using only the finetune set, with a single multi-session model trained across all finetune-set sessions and evaluated on the finetune test splits and zero-shot set; no additional pretraining is performed. These baselines test whether a conventional atlas-constrained decomposition, paired with either linear or contrastive decoding, is sufficient for cross-subject transfer.

**CEBRA.** CEBRA (Schneider et al., 2023) is a contrastive representation learning method that maps neural activity into behavior-informed latent embeddings. For the LocaNMF + CEBRA baseline, CEBRA is trained supervised using behavior labels from the finetune set only, while LocaNMF components are learned based on the sessions in the pretraining set. In the multi-session setting, a single CEBRA model is fit across all finetune-set sessions and evaluated on the finetune test splits and zero-shot set. For the decoder-fairness controls, we apply the MLP decoder and Kalman Filter/Smoother decoders to the CEBRA embeddings using the same setup as in Section F.2 and report the results in Table 8.

**NDT2.** We adapt NDT2 (Ye et al., 2023), a recent multi-subject model for electrophysiology, to widefield calcium imaging. To process neural images, we replace the spike embedding layer with a convolutional tokenizer that maps each $32 \times 32$ image patch (1024 pixels) to a token, and replace the original Poisson likelihood with an MSE loss for imaging data. Since NDT2 relies on session- and subject-specific embeddings, we include finetune-set sessions during NDT2 pretraining to learn their subject/session identifiers. Similar to WiCAT linear probing, we then train a linear decoder to predict behavior for each dataset. For zero-shot evaluation, we use subject/session identifiers from a random subject/session seen during pretraining; otherwise, we use the default NDT2 hyperparameters and settings.

**MLP.** We evaluate a supervised, single-session multilayer perceptron (MLP) baseline trained on flattened neural images. The MLP consists of two hidden layers with 1024 and 512 units, respectively.

**WiCAT (Single-Session).** To isolate the effect of multi-subject pretraining, we include a single-session variant of WiCAT trained independently for each session in the finetune set. In this baseline, the full WiCAT, including the tokenizer and Transformer backbone, is pretrained using the MAE objective on the neural data from a single session. A behavior decoder is then trained on top of the frozen backbone to decode behavior for that same session. This baseline mirrors standard single-session training in widefield imaging and assesses whether multi-subject pretraining provides benefits beyond single-session self-supervised training.

**WiCAT (JEPA Pretraining).** We compare against a JEPA-style (Assran et al., 2023) self-supervised pretraining variant inspired by Bardes et al. (2024). JEPA learns representations by predicting masked target embeddings from a context embedding using an asymmetric encoder-predictor architecture, where target embeddings are produced by a momentum-updated target network and predicted by the main network. In our experiments, we adapt JEPA to widefield calcium imaging by using the same atlas-aligned tokenization, global spatial embeddings, and Transformer backbone as WiCAT.

Following Bardes et al. (2024), we use a momentum update coefficient initialized at 0.998 and linearly annealed to 1.0 over the 500 epochs of pretraining. All other model architecture and optimization hyperparameters are set identically to those used for WiCAT.

We explored multiple masking strategies proposed in prior work, including block masking and random masking across space and time (Bardes et al., 2024; Dong et al., 2024) (See Appendix F.1). Despite extensive tuning (Table 6), JEPA-based pretraining did not outperform MAE for downstream behavior decoding in widefield data.

**WiCAT (Random Initialization).** In this variant, the full WiCAT architecture (backbone and decoder) is trained end-to-end from scratch on the finetune set without any self-supervised pretraining. This baseline isolates the effect of self-supervised pretraining from that of supervised optimization alone.

**WiCAT (Session-Specific Spatial Embeddings).** To evaluate the importance of global spatial embeddings, we introduce a variant that learns a separate set of spatial embeddings for each session. This model relies on session-specific parameters to encode spatial identity independently for each session. The session-specific positional embeddings are used by both

the encoder and the decoder. This ablation tests whether global (session-invariant) spatial embeddings are critical for cross-subject generalization.

**WiCAT (No Pos Enc).** This ablation removes global spatial embeddings entirely. All recordings are still aligned to the Allen Brain Atlas, but no explicit spatial identity is provided to the Transformer backbone. This variant tests whether spatial information must be explicitly encoded rather than inferred implicitly from image content alone.

## F. Additional Experiments and Figures

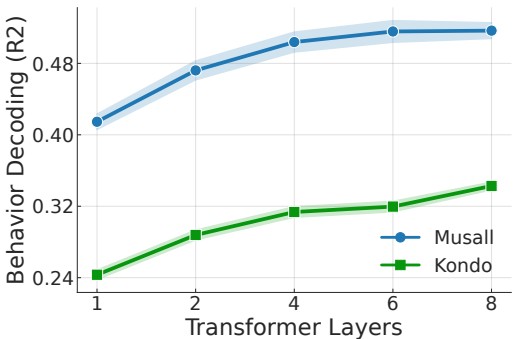

*Figure 9.* **Effect of model capacity on downstream behavior decoding performance.** Behavior decoding $R^2$ is shown as a function of the number of Transformer encoder layers for the Musall and Kondo datasets. Results are averaged over 3 finetuning seeds and all sessions (mean $\pm$ SEM). Performance on both datasets improves consistently with increasing model depth, with continued gains on the more challenging Kondo dataset. The Kondo dataset involves a more challenging and less temporally structured behavioral task (Appendix A), with lower average task success rates compared to Musall (Musall et al., 2019; Kondo et al., 2025), suggesting that additional model capacity remains beneficial for capturing the underlying neural-behavioral relationships. We use an 8-layer Transformer backbone in all main experiments as our default configuration. Model capacity increases with depth, corresponding to 6.6M (1 layer), 10.8M (2 layers), 19.2M (4 layers), 27.6M (6 layers), and 36.0M (8 layers) parameters.

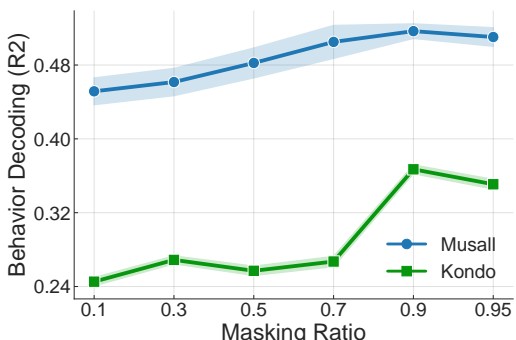

*Figure 10.* **Effect of masking ratio during self-supervised pretraining.** Downstream behavior decoding $R^2$ is shown for different masking ratios on the Musall and Kondo averaged over 3 finetuning seeds and all sessions (mean $\pm$ SEM). Performance peaks at a 90% masking ratio, indicating that aggressive masking is beneficial for learning richer representations for the downstream tasks. At this masking ratio, the encoder processes only 10% of the spatiotemporal tokens per batch, substantially reducing pretraining cost. Higher masking ratios (e.g., 95%) degrade downstream performance; we use a default 90% masking ratio in all main experiments.

### F.1. Self-Supervised Pretraining Masking Strategies

Several prior works have explored different masking strategies for masked self-supervised learning in spatiotemporal data, including masking entire temporal segments, spatial regions, or contiguous spatiotemporal blocks (Tong et al., 2022; Bardes et al., 2024; Dong et al., 2024; Oganesian et al., 2025). These strategies aim to encourage models to learn more informative context by making the pretraining task more challenging, especially that strong correlations exist between neighboring time points or spatial locations, or where specific structures are relevant for downstream tasks. Motivated by these approaches, we evaluate a range of masking strategies adapted to widefield calcium imaging within our self-supervised pretraining framework and atlas-aligned tokenization, to assess whether structured masking improves representation learning.

**Random token masking.** Our default strategy randomly masks individual spatiotemporal tokens across both space and time. This strategy applies masking uniformly over the flattened spatiotemporal token sequence.

**Frame masking.** Inspired by VideoMAE (Tong et al., 2022), we evaluated masking entire frames instead of individual spatiotemporal tokens. In this setting, all spatial patches corresponding to a given timestep in the recording segment are masked, such that 90% of timesteps are fully masked and reconstruction is performed from the remaining frames.

**Multi-block masking.** Following prior work on block masking (Bardes et al., 2024), we evaluated a multi-block masking strategy that masks contiguous spatiotemporal blocks. Specifically, for each recording segment, we randomly sample four spatiotemporal blocks of size $64 \times 64$ pixels and full temporal extent, and mask all tokens within these blocks. To ensure a minimum encoder context, we enforce that at least 10% of tokens remain unmasked by reintroducing randomly selected tubes if necessary.

**Causal multi-block masking.** We also evaluated a causal variant of multi-block masking (Bardes et al., 2024), where the context is restricted to the first portion of the sequence. In this setting, the final 40 timesteps of each segment are fully masked across all spatial locations, and reconstruction is performed using only earlier time points. As above, we enforce a minimum context ratio of 10% by unmasking additional tokens when required.

**Results.** As shown in Table 6, random token masking across space and time consistently yields the strongest downstream behavior decoding performance for both MAE and JEPA objectives. While structured masking strategies can be beneficial in other domains, we find that for widefield imaging data random masking combined with joint space-time attention is sufficient and performs best. We therefore adopt random token masking in all main experiments.

*Table 6.* **Effect of masking strategies for MAE and JEPA pretraining (Musall dataset, behavior decoding $R^2$ Mean $\pm$ SEM).** We evaluate several masking strategies adapted from prior work, including random token masking, frame (temporal) masking (Tong et al., 2022), multi-block masking, and causal multi-block masking (Bardes et al., 2024), for both MAE and JEPA objectives within our model. SEM computed across sessions and 5 finetuning seeds. Random token masking across space and time along with the spatiotemporal Transformer encoder architecture yields the strongest downstream decoding performance and is used in all main experiments. See Appendix F.1 for details of each masking strategy.

| | WiCAT (MAE) | | WiCAT (JEPA) | |
|---|---|---|---|---|
| **Masking Strategy** | **Finetune Set** | **Zero-shot Set** | **Finetune Set** | **Zero-shot Set** |
| **Random Token (Space + Time)** | $\mathbf{0.5124 \pm 0.0080}$ | $\mathbf{0.3274 \pm 0.0080}$ | $0.4987 \pm 0.0174$ | $0.2587 \pm 0.0330$ |
| **Frame Masking** | $0.4488 \pm 0.0092$ | $0.2547 \pm 0.0167$ | $0.4640 \pm 0.0166$ | $0.2607 \pm 0.0325$ |
| **Multi-Block Masking** | $0.4872 \pm 0.0099$ | $0.2859 \pm 0.0233$ | $0.4690 \pm 0.0142$ | $0.2000 \pm 0.0587$ |
| **Causal Multi-Block Masking** | $0.4341 \pm 0.0087$ | $0.2201 \pm 0.0153$ | $0.4459 \pm 0.0160$ | $0.1754 \pm 0.0414$ |

### F.2. Additional Downstream Decoder Details

For the additional decoder experiments, all model parameters and inferred representations remain frozen. The MLP decoder is a lightweight one-layer nonlinear decoder trained on the finetune set, with dropout probability 0.2 for the MLP-Dropout variant and no dropout for the MLP-NoDropout variant. For the Kalman decoder, let $h_t$ be the frozen encoder latent at time $t$, $x_t \in \mathbb{R}^{d_x}$ be the Kalman latent state with $d_x = 16$, and $z_t \in \mathbb{R}^{d_z}$ be the behavior target. We first project the encoder latent into the Kalman state space and then learn a linear dynamical system:

$$x_t = W_{\mathrm{proj}} h_t + b_{\mathrm{proj}}, \tag{1}$$

$$x_{t+1} = A x_t + w_t, \qquad w_t \sim \mathcal{N}(0, W), \tag{2}$$

$$z_t = C x_t + v_t, \qquad v_t \sim \mathcal{N}(0, R). \tag{3}$$

The learned parameters include $W_{\mathrm{proj}}$, $b_{\mathrm{proj}}$, the initial state mean $\mu_0$, initial covariance $\Lambda_0$, transition matrix $A$, readout matrix $C$, and diagonal process and observation noise covariances parameterized as $W = \mathrm{diag}(\exp(w_{\mathrm{log}}))$ and $R = \mathrm{diag}(\exp(r_{\mathrm{log}}))$. The causal variant uses the forward filtered estimate, whereas the smoothing variant uses full-sequence

smoothed states. Table 8 reports controls in which nonlinear or dynamical decoders are also applied to the learned representations from LocaNMF + CEBRA, NDT2, and SBIND baselines.

*Table 7.* **Additional baseline comparisons including adapted NDT2 and LocaNMF + CEBRA.** Behavior decoding $R^2$ (mean $\pm$ SEM), averaged across sessions. NDT2 and CEBRA were adapted to the widefield calcium imaging setting as described in Appendix E.

| Model | Musall Behavior Decoding $R^2 \uparrow$ | | Kondo Behavior Decoding $R^2 \uparrow$ | |
| --- | --- | --- | --- | --- |
| | **Finetune Set** | **Zero-shot Set** | **Finetune Set** | **Zero-shot Set** |
| WiCAT (Linear Probing) | $0.5124 \pm 0.0080$ | $0.3274 \pm 0.0080$ | $0.3396 \pm 0.0038$ | $0.1840 \pm 0.0053$ |
| NDT2 (Linear Probing) | $0.4775 \pm 0.0201$ | $0.1296 \pm 0.0146$ | $0.2467 \pm 0.0098$ | $0.0698 \pm 0.0126$ |
| LocaNMF + CEBRA | $0.4595 \pm 0.0084$ | $0.2421 \pm 0.0334$ | $0.2417 \pm 0.0079$ | $0.0853 \pm 0.0082$ |

### F.3. Chance-Level Baselines

We compute chance-level behavior decoding baselines using 1000 permutation repetitions. Within each repetition, we keep neural activity fixed and shuffle behavior targets within each finetune-train session, breaking neural-behavioral alignment while preserving the target statistics used by the shuffled fit. We then fit a multi-session *PCA + linear regression* baseline on the shuffled finetune-train data and evaluate the fitted model on the test split of both the finetune and zero-shot sets. Significance is assessed with a one-sided permutation test within each session, comparing the WiCAT decoding score to the shuffled scores obtained using this procedure.

For neural reconstruction, we compute chance-level baselines similarly but using 10 permutations. For each target ROI, we keep the source brain regions fixed and shuffle trial-length blocks of target ROI activity across trials within each session. We then fit the same linear decoder baseline used in the main reconstruction experiments to predict the shuffled target ROI activity from the remaining brain activity. Significance is assessed with a one-sided Wilcoxon signed-rank test across sessions.

This yields a conservative chance-level baseline for behavior decoding because shuffled Musall trials can still share the same task-aligned time axis and trial structure. In contrast, Kondo behavior is less constrained and less consistently aligned across trials, so the same shuffle produces near-zero chance levels. Behavior decoding chance-level results are reported in Table 10; neural reconstruction chance-level results are reported in Table 11.

### F.4. Atlas Perturbation Analysis

Because WiCAT uses atlas alignment to define a shared spatial coordinate system, we evaluate sensitivity to controlled affine perturbations. For each session, we either keep the original atlas alignment unchanged or sample a random affine perturbation independently for that session. Perturbed sessions use random scaling in the range 0.9–1.1, pixel shifts of up to $\pm 5$ pixels, and random rotations whose maximum absolute angle is specified by each row of Table 12. Small affine perturbations have limited effect, while larger rotations degrade zero-shot decoding, indicating that WiCAT is robust to minor registration noise but relies on reasonable atlas alignment for cross-subject transfer.

*Table 8.* **Downstream decoder comparison across models.** Behavior decoding $R^2$ (mean $\pm$ SEM) for zero-shot sets. All backbones are frozen and decoders are trained on the finetune set. NDT2 and CEBRA are adapted to widefield imaging as described in Appendix E. Results for SBIND are for a multi-session extension of the original method in Hosseini & Shanechi (2025).

| Model | Musall Zero-shot Set | Kondo Zero-shot Set |
|---|---|---|
| WiCAT + Linear | $0.3274 \pm 0.0080$ | $0.1840 \pm 0.0053$ |
| WiCAT + Kalman Filter | $0.4313 \pm 0.0055$ | $0.2743 \pm 0.0053$ |
| WiCAT + Kalman Smoother | $\mathbf{0.4336 \pm 0.0056}$ | $\mathbf{0.2869 \pm 0.0054}$ |
| WiCAT + MLP | $0.4115 \pm 0.0229$ | $0.2867 \pm 0.0115$ |
| NDT2 + Linear | $0.1296 \pm 0.0146$ | $0.0698 \pm 0.0126$ |
| NDT2 + Kalman Filter | $0.1216 \pm 0.0143$ | $0.0662 \pm 0.0080$ |
| NDT2 + Kalman Smoother | $0.1624 \pm 0.0140$ | $0.0967 \pm 0.0081$ |
| NDT2 + MLP | $0.1584 \pm 0.0234$ | $0.0995 \pm 0.0093$ |
| LocaNMF + CEBRA + Linear | $0.2421 \pm 0.0334$ | $0.0853 \pm 0.0082$ |
| LocaNMF + CEBRA + Kalman Filter | $0.2316 \pm 0.0389$ | $0.0513 \pm 0.0077$ |
| LocaNMF + CEBRA + Kalman Smoother | $0.2524 \pm 0.0309$ | $0.0879 \pm 0.0076$ |
| LocaNMF + CEBRA + MLP | $0.2462 \pm 0.0391$ | $0.0302 \pm 0.0060$ |
| LocaNMF + Linear Regression | $0.1991 \pm 0.0303$ | $0.0948 \pm 0.0074$ |
| SBIND-Unsup + Kalman Filter | $0.2591 \pm 0.0321$ | $0.1512 \pm 0.0075$ |
| SBIND-Unsup + Kalman Smoother | $0.2751 \pm 0.0406$ | $0.1665 \pm 0.0091$ |
| SBIND-Sup + Kalman Smoother | $0.2783 \pm 0.0311$ | $0.1615 \pm 0.0096$ |

*Table 9.* **Detailed zero-shot neural reconstruction error (pixel-wise MSE, Mean ± SEM) for left-out brain *subregions*.** Rows report atlas-defined subregions grouped by their parent cortical area (group headers shown in bold). For each subregion, MSE is computed by averaging pixel-wise reconstruction error over all pixels belonging to that subregion, and then averaging across sessions (SEM is computed across sessions). ROI-level results reported in Table 4 are computed independently and are therefore *not* simple averages of the corresponding subregion values. The **Average** reports the mean MSE aggregated across 44 atlas-defined subregions and 168 sessions. Results are shown for WiCAT and a Linear Decoder baseline.

| ROI Group / Subregion | WiCAT ↓ | Linear Decoder ↓ |
|---|---|---|
| **SS** | | |
| SSp_bfd_l | 0.3147 ± 0.0119 | 0.5308 ± 0.0321 |
| SSp_bfd_r | 0.2726 ± 0.0095 | 0.3269 ± 0.0080 |
| SSp_ll_l | 0.6026 ± 0.0211 | 2.7006 ± 0.1411 |
| SSp_ll_r | 0.7390 ± 0.0329 | 2.3628 ± 0.1235 |
| SSp_m_l | 0.0845 ± 0.0047 | 0.1746 ± 0.0094 |
| SSp_m_r | 0.0607 ± 0.0041 | 0.1085 ± 0.0041 |
| SSp_n_l | 0.1795 ± 0.0065 | 0.3802 ± 0.0222 |
| SSp_n_r | 0.1263 ± 0.0061 | 0.2370 ± 0.0113 |
| SSp_tr_l | 0.6419 ± 0.0163 | 1.6401 ± 0.0931 |
| SSp_tr_r | 0.6423 ± 0.0189 | 1.4595 ± 0.0676 |
| SSp_ul_l | 0.2766 ± 0.0085 | 1.3065 ± 0.0708 |
| SSp_ul_r | 0.2560 ± 0.0114 | 0.9959 ± 0.0422 |
| SSp_un_l | 0.4512 ± 0.0157 | 1.0856 ± 0.0725 |
| SSp_un_r | 0.3923 ± 0.0158 | 0.6670 ± 0.0222 |
| SSs_l | 0.0258 ± 0.0016 | 0.0277 ± 0.0018 |
| SSs_r | 0.0170 ± 0.0011 | 0.0143 ± 0.0007 |
| **VIS** | | |
| VISa_l | 0.9442 ± 0.0456 | 0.9802 ± 0.0413 |
| VISa_r | 0.7232 ± 0.0274 | 0.7983 ± 0.0239 |
| VISal_l | 0.3506 ± 0.0281 | 0.3580 ± 0.0298 |
| VISal_r | 0.1735 ± 0.0129 | 0.1802 ± 0.0116 |
| VISam_l | 0.9477 ± 0.0460 | 0.8918 ± 0.0441 |
| VISam_r | 0.7451 ± 0.0321 | 0.7332 ± 0.0290 |
| VISl_l | 0.0527 ± 0.0034 | 0.0540 ± 0.0038 |
| VISl_r | 0.0213 ± 0.0016 | 0.0225 ± 0.0016 |
| VISp_l | 0.5813 ± 0.0302 | 0.6868 ± 0.0396 |
| VISp_r | 0.3755 ± 0.0136 | 0.4268 ± 0.0148 |
| VISpm_l | 0.7907 ± 0.0392 | 0.8954 ± 0.0519 |
| VISpm_r | 0.6181 ± 0.0265 | 0.6788 ± 0.0291 |
| VISrl_l | 1.0273 ± 0.0737 | 1.0705 ± 0.0748 |
| VISrl_r | 0.5775 ± 0.0285 | 0.6271 ± 0.0256 |
| **MO** | | |
| MOp_l | 0.4042 ± 0.0124 | 0.3598 ± 0.0095 |
| MOp_r | 0.2923 ± 0.0078 | 0.3024 ± 0.0106 |
| MOs_l | 0.3062 ± 0.0110 | 0.3263 ± 0.0158 |
| MOs_r | 0.2114 ± 0.0063 | 0.2537 ± 0.0138 |
| **AUD** | | |
| AUDp_l | 0.0169 ± 0.0010 | 0.0161 ± 0.0011 |
| AUDp_r | 0.0097 ± 0.0006 | 0.0088 ± 0.0007 |
| AUDs_l | 0.0385 ± 0.0024 | 0.0391 ± 0.0030 |
| AUDs_r | 0.0210 ± 0.0011 | 0.0187 ± 0.0010 |
| **OB** | | |
| OB_l | 0.0356 ± 0.0021 | 0.0519 ± 0.0026 |
| OB_r | 0.0238 ± 0.0025 | 0.0382 ± 0.0028 |
| **RSP** | | |
| RSPagl_l | 0.2283 ± 0.0076 | 0.2100 ± 0.0066 |
| RSPagl_r | 0.2153 ± 0.0066 | 0.2223 ± 0.0078 |
| RSPd_l | 0.1934 ± 0.0071 | 0.2142 ± 0.0084 |
| RSPd_r | 0.1861 ± 0.0071 | 0.2044 ± 0.0082 |
| **Average (All Subregions)** | **0.3453 ± 0.0047** | 0.5611 ± 0.0095 |

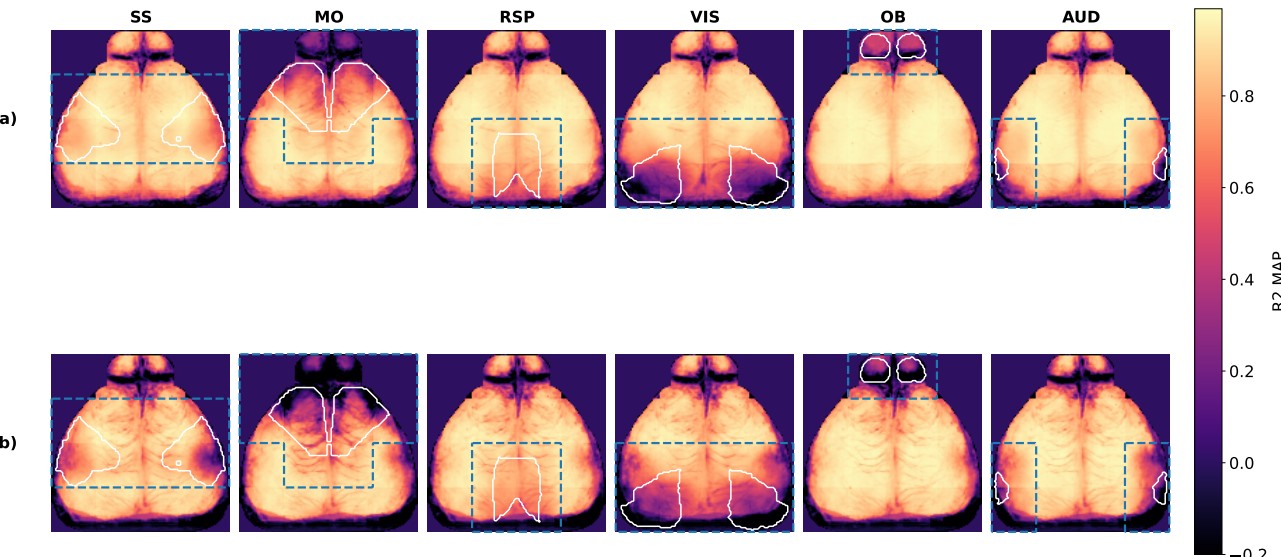

*Figure 11.* **Per-pixel $R^2$ maps for left-out region prediction.** Per-pixel $R^2$ for **(a)** held-in subjects (top row; pretraining set, sessions seen during pretraining) and **(b)** held-out subjects (bottom row; finetune and zero-shot sets, with no finetuning performed for this reconstruction analysis), averaged across all sessions. For each masked ROI column, the top panel is averaged across all pretraining sessions, while the bottom panel is averaged across all held-out (finetune and zero-shot set) sessions together; see Appendix B.5 for more details on the evaluation setup. Each column corresponds to masking the region bounded by the blue dashed lines and predicting its activity from the rest of the brain as context. The white boundaries in each panel illustrate the target ROI. Evaluation for each column focuses on the corresponding masked ROI, where the ROI and its neighboring regions are removed to eliminate local spatial correlations, ensuring that performance reflects cross-region prediction rather than trivial neighborhood interpolation. A higher $R^2$ within the masked region indicates successful cross-region prediction. Darker maps in regions such as AUD and OB appear in both held-in and held-out subjects, indicating lower explained variance due to very low neural variance and higher sensitivity to noise rather than a failure of zero-shot generalization; these regions are also near atlas boundaries, where small misalignments can lead to larger errors. The results show that WiCAT maintains strong predictive performance across both seen and unseen subjects, with performance largely preserved when transitioning to held-out subjects, indicating that global spatial patterns are learned and transferred effectively to the zero-shot set.

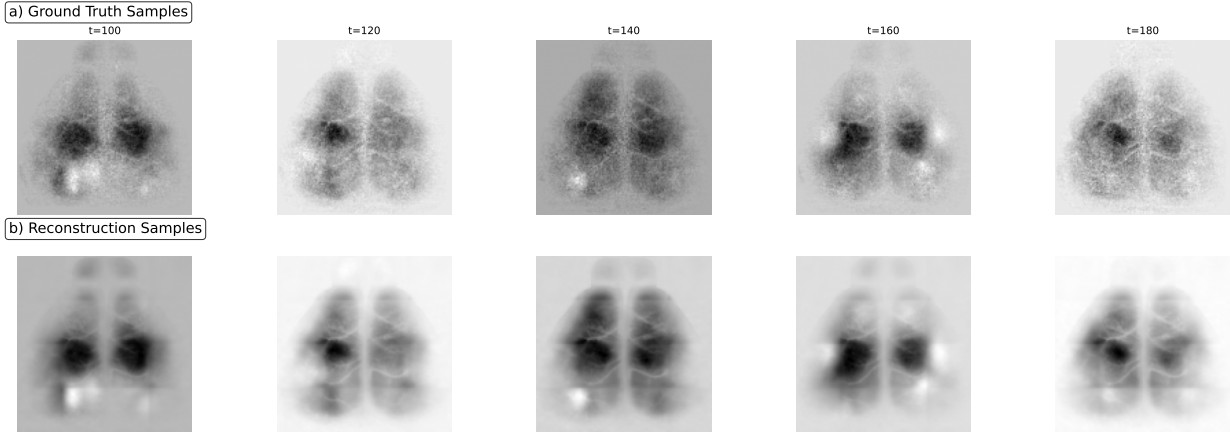

*Figure 12.* **Qualitative snapshots of neural reconstruction with masked regions.** Snapshots of **(a)** widefield activity (top row) across multiple time frames compared to **(b)** model predictions (bottom row). In these examples, all visual area (VIS) patches are masked (as indicated by the blue bounding boxes in Fig. 11, column VIS). Results are shown for a representative zero-shot (held-out) subject. Despite this, WiCAT accurately reconstructs the missing activity, demonstrating its ability to perform cross-region neural prediction from the available context.

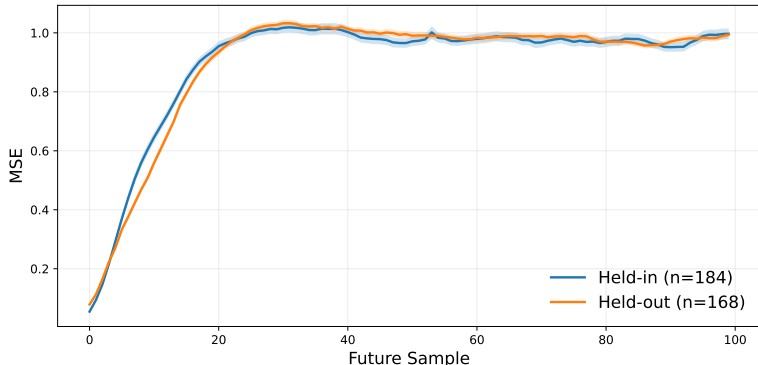

*Figure 13.* **Temporal prediction without future context.** Mean squared error (MSE) across all pixels for predicting future frames when only the first part of the trial is provided as input (last 100 frames predicted without context), evaluated across held-in (pretraining set) and held-out (finetune and zero-shot sets) subjects. The results show that WiCAT preserves temporal structure and remains robust when generalizing to unseen subjects.

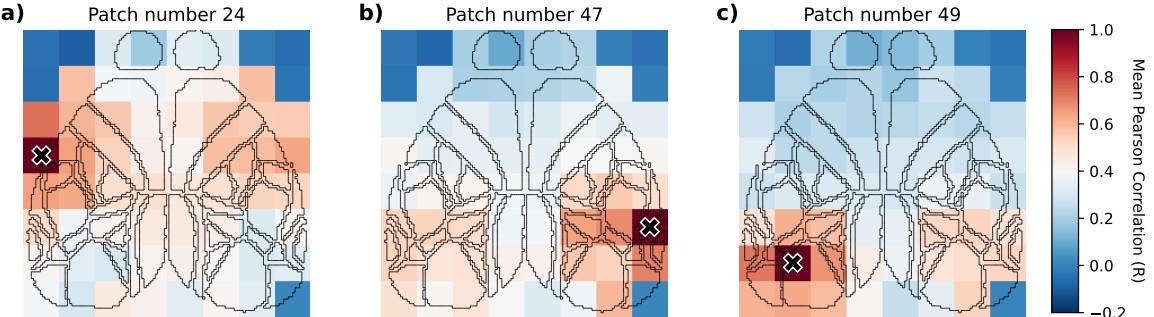

*Figure 14.* **Correlation structure of learned patch embeddings.** Mean pairwise correlations between patch embeddings, aggregated across sessions. Each panel shows, as a heatmap, the correlation of all other patches with a reference patch: **(a)** 24, **(b)** 47, and **(c)** 49. The reference patch is marked, and Allen atlas boundaries are overlaid. Correlations are strongest for neighboring regions and homologous areas across hemispheres. This structured correlation pattern further shows that the learned representations capture spatially coherent and anatomically meaningful relationships.

*Table 10.* **Chance-level behavior decoding** ($R^2$, mean $\pm$ SEM over sessions). Chance is computed using the permutation procedure described in Appendix F.3: over 1000 repetitions, behavior targets are shuffled within finetune-train sessions, the *PCA+linear regression* decoder is refit, and the fitted model is evaluated on finetune set and zero-shot set sessions. The lower absolute $R^2$ values on Kondo in Table 1 reflect its less temporally structured lever-pull task and higher session variability as further analyzed in Figure 15, whereas the task-aligned Musall trials yield a conservative positive chance level. Across all sessions, WiCAT is significantly above chance (one-sided permutation test, $N = 1000$, $p < 0.001$).

| Dataset | Split | WiCAT $R^2$ | Chance $R^2$ | $N$ |
|---|---|---|---|---|
| Musall | Finetune Set | $0.5124 \pm 0.0080$ | $0.2348 \pm 0.0202$ | 6 |
| Musall | Zero-shot Set | $0.3274 \pm 0.0080$ | $0.1133 \pm 0.0436$ | 6 |
| Kondo | Finetune Set | $0.3396 \pm 0.0038$ | $-0.0009 \pm 0.0002$ | 87 |
| Kondo | Zero-shot Set | $0.1840 \pm 0.0053$ | $-0.0034 \pm 0.0003$ | 81 |

*Table 11.* **Chance-level zero-shot neural reconstruction** on Kondo (pixel-wise MSE, mean $\pm$ SEM over $N = 168$ sessions). Chance is computed using the permutation procedure described in Appendix F.3: for each target ROI, source brain regions are fixed while trial-length blocks of target ROI activity are shuffled across trials within each session over 10 permutations, and the same linear decoder baseline is fit to predict the shuffled target activity. Lower MSE is better; WiCAT is significantly better than chance for all ROIs (one-sided Wilcoxon signed-rank test, $n = 168$, $p < 0.001$ for all regions).

| ROI | WiCAT MSE | Chance MSE |
|-----|-----------|------------|
| SS | $0.2914 \pm 0.0065$ | $2.4355 \pm 0.0497$ |
| VIS | $0.4963 \pm 0.0196$ | $1.1381 \pm 0.0489$ |
| MO | $0.2932 \pm 0.0070$ | $1.0944 \pm 0.0261$ |
| AUD | $0.0219 \pm 0.0008$ | $0.0566 \pm 0.0038$ |
| OB | $0.0298 \pm 0.0021$ | $0.0630 \pm 0.0066$ |
| RSP | $0.1990 \pm 0.0066$ | $1.1175 \pm 0.0238$ |

*Table 12.* **Atlas perturbation effect on behavior decoding** ($R^2$). Rows report affine perturbations applied to the atlas alignment. Perturbed rows jointly include random scaling in the range 0.9–1.1 and pixel shifts of up to $\pm 5$ pixels; the table row reports the maximum absolute rotation angle sampled for each session.

| Dataset | Max Rotation | Finetune Set | Zero-shot Set |
|---------|--------------|--------------|---------------|
| Musall | No perturb | $0.5124 \pm 0.0080$ | $0.3274 \pm 0.0080$ |
| Musall | $0°$ | $0.4911 \pm 0.0165$ | $0.3318 \pm 0.0149$ |
| Musall | $30°$ | $0.4477 \pm 0.0133$ | $0.1342 \pm 0.0732$ |
| Kondo | No perturb | $0.3396 \pm 0.0038$ | $0.1840 \pm 0.0053$ |
| Kondo | $10°$ | $0.2959 \pm 0.0079$ | $0.1290 \pm 0.0110$ |
| Kondo | $30°$ | $0.2873 \pm 0.0079$ | $0.1384 \pm 0.0110$ |

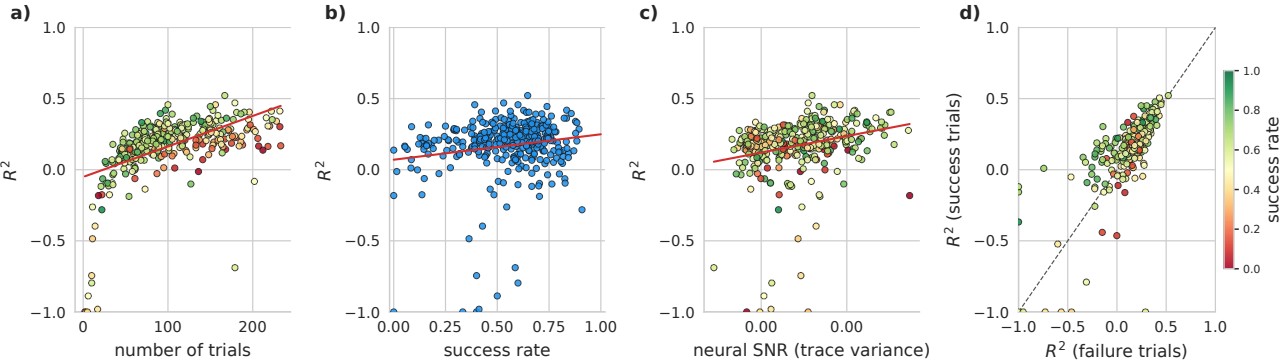

*Figure 15.* **Session-level factors affecting decoding performance on Kondo.** **(a)** $R^2$ vs. number of trials per session (color indicates success rate), **(b)** $R^2$ vs. session success rate, reflecting behavioral engagement, **(c)** $R^2$ vs. neural SNR (mean PC trace variance from PCA), and **(d)** $R^2$ for successful vs. failed trials (color indicates success rate; dashed line is identity, so points above the line indicate better decoding). Across sessions ($n = 352$), trial count ($r = 0.52$, $p < 0.001$), neural SNR proxy ($r = 0.246$, $p < 0.001$), and success rate as a proxy for task engagement ($r = 0.17$, $p = 0.002$) positively correlate with decoding performance as seen in panels (a)–(c). Sessions with higher engagement cluster above the identity line in panel (d), indicating that decoding is easier during successful trials.

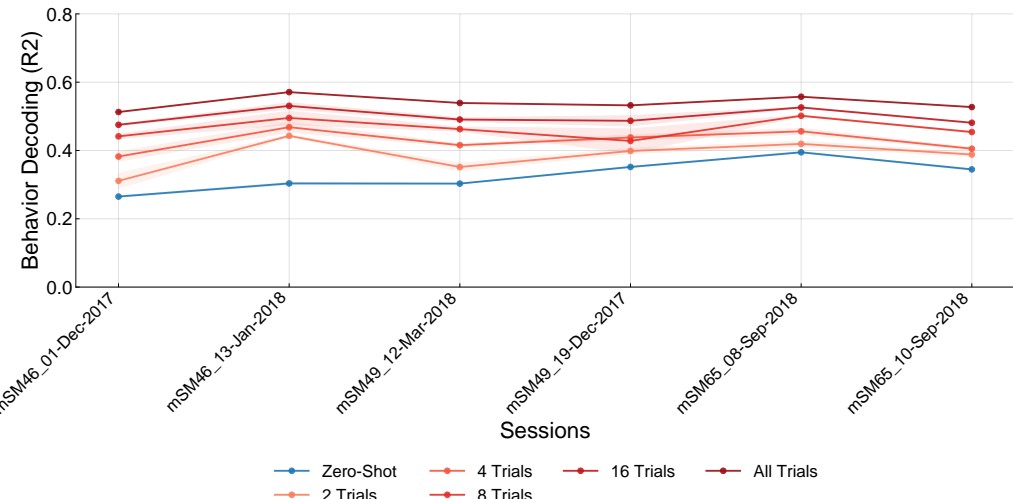

*Figure 16.* **Per-session behavior decoding performance on the Musall task dataset.** $R^2$ is shown for zero-shot, few-shot (2, 4, 8, 16 trials), and full-shot settings, averaged over 5 finetuning seeds. Decoding performance improves consistently as additional labeled trials are used for adaptation. Error bars indicate SEM across seeds.

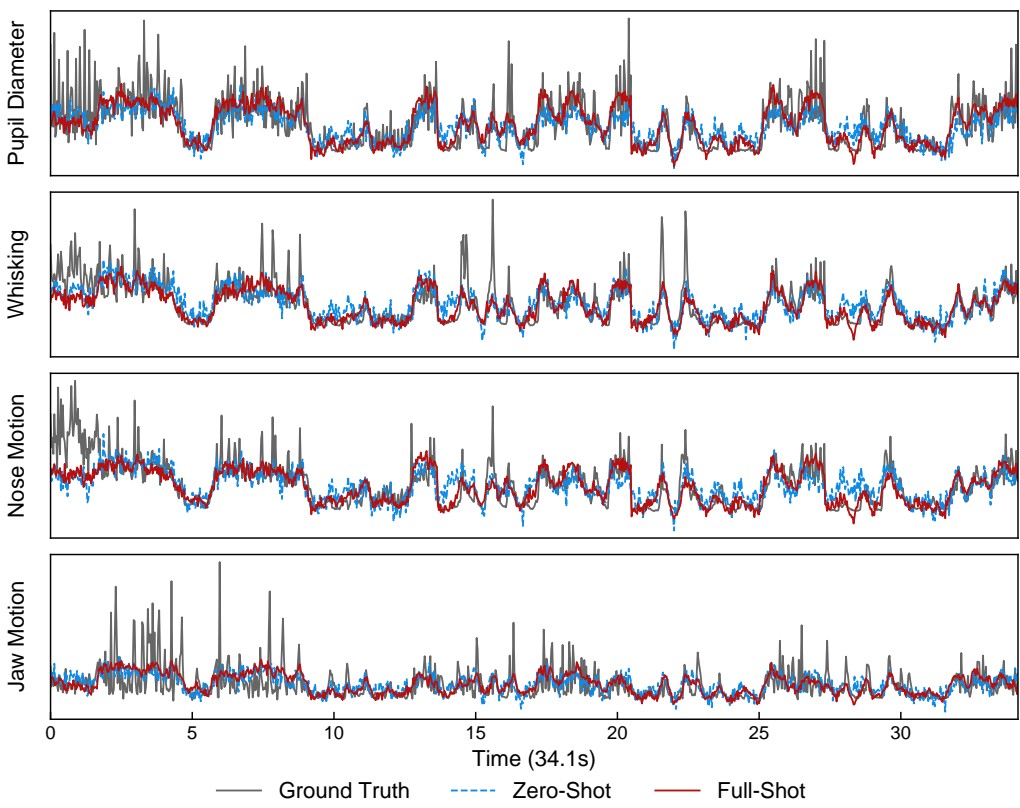

*Figure 17.* **Behavior decoding traces on the Musall dataset.** Predicted behavior is shown over 34.1 seconds (5 trials) for zero-shot decoding using the frozen decoder and full-shot finetuned decoder trained on a representative session. Both zero-shot and full-shot predictions capture the overall temporal trends in the behavioral signals, while full-shot decoding more accurately matches the ground-truth.

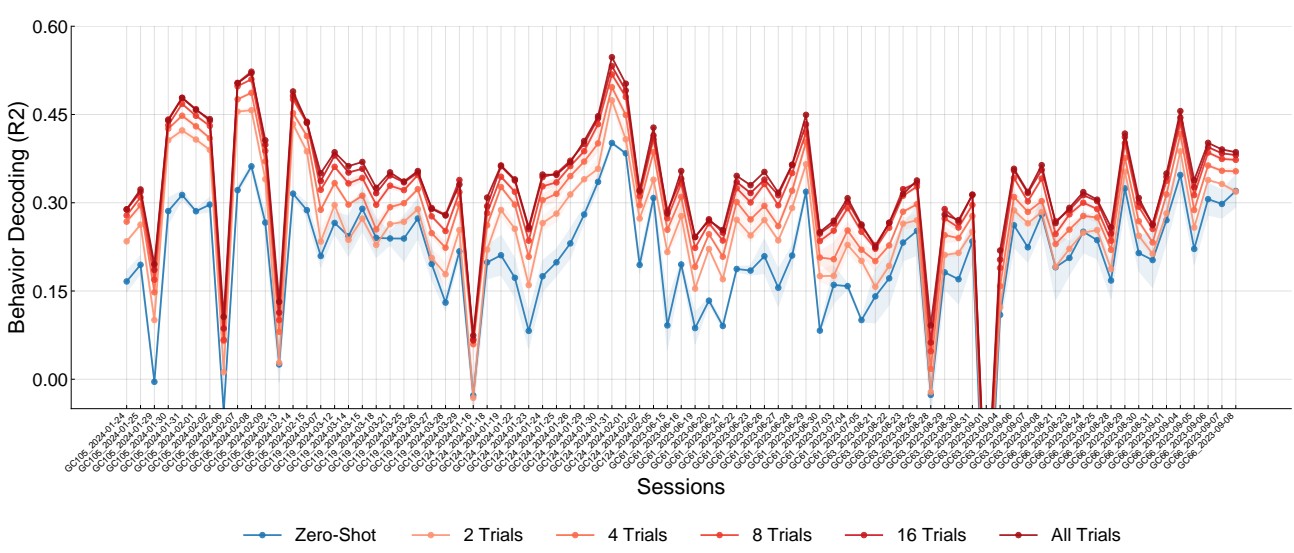

*Figure 18.* **Per-session behavior decoding performance on the Kondo dataset.** Figure conventions are the same as in Figure 16.

![Behavior decoding traces with labels lowerjaw, nosebottom, noseroot, earlateral, eartip, pupiledge01 over Time (35.0s)]

Legend: Ground Truth, Zero-Shot, Full-Shot

*Figure 19.* **Behavior decoding traces on the Kondo dataset.** Predicted behavior is shown over 35.0 seconds (5 trials) for zero-shot decoding and full-shot linear decoder trained on a representative session. Figure conventions are the same as in Figure 17.

