# OpenReview forum: "Cross-Subject Modeling for Widefield Calcium Imaging via Atlas-Aligned Spatiotemporal Tokenization"
_ICML.cc/2026/Conference — ICML 2026 regular_

### Official Review · Reviewer_YxAF · 2026-03-11

**Soundness:** 3
**Presentation:** 3
**Significance:** 3
**Originality:** 3
**Overall Recommendation:** 4
**Confidence:** 3

**Summary:**

This paper presents WiCAT, a model for widefield calcium imaging that learns shared spatiotemporal representations across subjects and sessions using self-supervised pretraining. The method has three main components. First, atlas-grounded tokenization registers all recordings to the Allen Brain Atlas and divides each frame into non-overlapping spatiotemporal patches. Second, global spatial embeddings encode anatomical identity and are shared across all subjects and sessions without introducing session-specific parameters. Third, a Transformer backbone is pretrained with a masked autoencoding (MAE) objective at a 90% masking ratio. The authors evaluate the model on two publicly available datasets: Musall (13 mice, 26 sessions, visual decision-making) and Kondo (25 mice, 352 sessions, auditory lever-pull). WiCAT achieves competitive performance on finetuned subjects, zero-shot decoding on unseen subjects, cross-dataset transfer, left-out brain region reconstruction, and shows favorable scaling with data size and model capacity.

**Compliance With Llm Reviewing Policy:**

Affirmed.

**Final Justification:**

I thank the authors for their effort across two rounds of revision. Several concerns have been effectively addressed: the decoder-fairness comparison (Table 12), the noise ceiling analysis, the removal of "foundation model" terminology, and the region-wise variance statistics have all improved the paper.

However, concerns remain: (1) the noise ceiling analysis relies on assumptions that the authors themselves acknowledge may not hold, making it only an approximation; (2) the experimental scale is limited (2 datasets, 38 mice, head-fixed paradigms only), constraining the generality of the conclusions; (3) the registration robustness analysis remains incomplete, lacking scaling/shift perturbations and results on the Kondo dataset.

Overall, this is solid work and the revisions have improved the paper, but the unresolved issues lead me to maintain my original score.

**Key Questions For Authors:**

1. The zero-shot R² of 0.18 on the Kondo dataset is modest. Can the authors provide context for what R² values are theoretically achievable given the signal-to-noise ratio and neural variability in widefield calcium imaging? This would help assess whether 0.18 represents a meaningful decoding result.
2. Have the authors evaluated lightweight nonlinear decoders, such as a 1–2 layer MLP on frozen representations? The gap between linear probing and full finetuning suggests nonlinear structure in the representations. This could help separate decoder limitations from representation quality.
3. In Table 3, the AUD region shows slightly better performance with a linear decoder than with WiCAT. Do the authors have an explanation for this? An analysis of which regions benefit most from cross-region modeling would be informative.
4. The approach depends on accurate atlas registration. How sensitive are the results to the quality of atlas alignment? Have experiments been performed with degraded registration (e.g., random affine perturbations)?
5. Figures 10 and 12 show substantial per-session variance. Can the authors analyze factors that predict zero-shot decoding quality, such as session SNR, behavioral engagement, trial count, or distance of cortical window from atlas regions? Understanding these factors would guide practical use.

**Limitations:**

The discussion acknowledges some limitations, such as the use of linear decoders and the need for larger datasets, but several important points are not addressed. The evaluation is limited to head-fixed mice performing motor-related tasks, leaving generalizability to other behaviors, species, or imaging configurations untested. The dependence on atlas registration quality is not discussed, even though it is critical to the framework. The modest absolute zero-shot R² values, particularly 0.18 on Kondo, are not contextualized in terms of practical scientific utility. Finally, the paper does not discuss potential failure modes or conditions under which cross-subject transfer might fail, such as unusual cortical morphology, developmental differences, or pathological conditions.

**Strengths And Weaknesses:**

Strengths
1. WiCAT addresses an important gap in widefield calcium imaging by providing the first model that can learn shared representations across multiple subjects and sessions. The experimental design is rigorous, with fully disjoint pretrain, finetune, and zero-shot subject sets, ensuring that reported performance reflects genuine generalization rather than data leakage.
2. The authors conduct comprehensive ablation studies, systematically isolating the contributions of global spatial embeddings, pretraining, and the choice of objective function, and they further explore masking strategies and ratios.
3. Zero-shot continuous behavior decoding on unseen subjects is statistically validated, and few-shot adaptation demonstrates rapid convergence with minimal data. Cross-dataset transfer experiments and scaling analyses show that the learned representations generalize to different tasks and benefit from increased data volume and model capacity.
4. Both evaluation datasets are publicly available, and preprocessing steps, hyperparameters, and model specifications are provided, supporting reproducibility.

Weaknesses
1. The zero-shot performance on the Kondo dataset is modest, with R² values of only 0.18, which limits practical interpretability. The evaluation is also restricted to head-fixed mice performing motor-related tasks with widefield dorsal cortex imaging, so claims of foundation-style generalization are not fully supported.
2. The pretraining dataset is relatively small compared to large foundation models, and the paper should acknowledge the gap between current scale and what would constitute a true foundation model.
3. All downstream behavior decoding experiments use linear decoders on frozen representations, which may underestimate the utility of the learned embeddings. Neural reconstruction results are region-dependent, with the AUD region showing better performance with a linear decoder, but this exception is not discussed.
4. There is no comparison with general neural foundation models adapted to widefield imaging, such as NDT2, POYO, or CEBRA. Error analysis and discussion of failure modes are missing, and computational cost comparisons with baselines are not provided.

---

> ### Author Rebuttal · Authors · 2026-03-31
>
> We thank the reviewer for their constructive suggestions for new analyses, which improve our work, and for recognizing the important gap in widefield imaging addressed here.
>
> ---
> ## [W1, Q1] Zero-shot (ZS) Interpretation
> We refer the reviewer to our response to Reviewer 9vGp (W1) for a discussion of ZS results and interpretation.
>
> ---
> ## [W2] Data & Task Scale
> We refer the reviewer to our response to Reviewer 9vGp (W2) for discussion of dataset scale and diversity. As explained there, we will add that the current available widefield data scale is smaller than in other modalities as a limitation to the Discussion section.
>
> We also agree that the tasks are limited to head-fixed mice and will clarify this in the Discussion. However, this reflects the current state of the field: most widefield studies are conducted in head-fixed settings due to hardware constraints. While recent work has begun extending imaging to freely moving animals (e.g., Kocek et al., 2021; Zhang et al., 2024), these approaches face significant trade-offs in field of view, resolution, and SNR, and publicly available datasets in this regime remain limited. As a result, head-fixed datasets remain the majority of available datasets (Hu et al., 2025). Nevertheless, the tasks considered include visual and auditory stimuli as well as decision-making and movement, providing a diverse testbed within the head-fixed setting.
>
> Kocek, et al., High-speed volumetric imaging of neuronal activity in freely moving rodents
>
> Hu et al., Pan-cortical cellular imaging in freely behaving mice using mini-MCAM
>
> Zhang et al., A miniaturized mesoscope for the large-scale single-neuron-resolved imaging of neuronal activity
>
> ---
> ## [W3, Q2] Representation Quality & Decoders
> We agree that linear decoders may underestimate the quality of the learned representations. We now evaluate additional lightweight decoders (**new Table 10**), including a 1-layer MLP and Kalman filter variants (dynamic). For the Kalman decoder, we first apply a linear layer to project the embeddings to a 16-dimensional latent space, treated as Kalman latent states, and behavior is predicted from these states (parameters estimated via numerical optimization). These improve ZS performance (e.g., from R² = 0.18 to 0.29 on Kondo), indicating that learned representations contain predictive structure beyond what was revealed by simple linear probing (LP).
>
> **Table 10. Downstream Decoder Comparisons.** Same convention as Table 1. * indicates significantly better than others (Wilcoxon signed-rank, p<0.005, n=sessions×seeds). † indicates frozen WiCAT model during downstream decoding.
>
> WiCAT Decoder|Musall FS|Musall ZS|Kondo FS|Kondo ZS
> ---|---|---|---|---
> Linear†|0.51±0.01|0.33±0.01|0.34±0.00|0.18±0.01
> KalmanCausal†|0.51±0.01|0.43±0.01|0.33±0.00|0.27±0.01
> KalmanSmooth†|0.51±0.01|0.43±0.01*|0.34±0.00|0.29±0.01*
> MLP+Dropout(0.2)†|0.54±0.02|0.41±0.02|0.36±0.01|0.29±0.01
> MLP+NoDropout†|0.56±0.02|0.37±0.03|0.37±0.03|0.24±0.01
> Full FT+Linear|0.58±0.01*|0.37±0.01|0.42±0.01*|0.24±0.01
>
> ---
> ## [W4] Baseline Comparisons
> We now add additional suggested baselines (used for other neural modalities), including NDT2 and CEBRA (**new Table 7**; please see response to Reviewer 9vGp, W3). WiCAT consistently outperforms these methods, supporting the benefits of self-supervised cross-subject representation learning.
>
> Computation cost: NDT2 pretraining requires ~2× more time than WiCAT. For downstream tasks, LP is efficient, while inferring the embeddings from the pretrained model is the main cost.
>
> ---
> ## [Q3] Performance in AUD Regions
> The AUD areas exhibit very low variance compared to other brain regions, suggesting they are not as active/relevant in these datasets, making predictions more sensitive to noise. Furthermore, AUD regions are located near the atlas boundaries. We speculate that small misalignments in ZS subjects can lead to larger errors in these regions.
>
> ---
> ## [Q4] Sensitivity to Registration
> We investigate the effect of atlas misalignment via affine perturbations, including scaling (0.9-1.1), shifts (±5 pixels), and rotations (up to 30°), on Musall dataset. Results (**new Table 11**) show minimal impact under small perturbations, while larger rotations significantly degrade ZS decoding. This indicates that WiCAT is robust to minor registration errors but relies on reasonable alignment for cross-subject generalization, and that transfer may degrade under substantial misalignment or anatomical variability, which we will add to the limitations.
>
> **Table 11. Atlas Perturbation Effect on Decoding R²**
> |Perturb/Rotation|FT|ZS|
> -|-|-
> No perturb|0.51±0.01|0.33±0.01
> 10°|0.49±0.02|0.32±0.01
> 30°|0.45±0.01|0.13±0.06
>
> ---
> ## [Q5] Dataset Quality
> We agree that session-level variability affects decoding performance. As shown in **new Fig. 19** (see here: https://anonymous.4open.science/r/WiCAT/Figs.pdf), trial count, behavioral engagement (trial success rate), and neural SNR all positively correlate with performance.

---

> > ### Author Rebuttal · Reviewer_YxAF · 2026-04-02
> >
> > I thank the authors for their rebuttal and the additional experiments. Several concerns have been addressed, but some issues remain and new questions have emerged from the added results.
> >
> > **Addressed concerns:**
> >
> > [W4]: The addition of NDT2 and CEBRA baselines is appreciated. [Q4]: The registration sensitivity analysis (Table 11) provides useful practical guidance. [Q5]: The analysis of factors predicting ZS decoding quality is a valuable addition.
> >
> > **Remaining concerns and new questions:**
> >
> > 1. **[W3, Q2] and Table 10**: The new decoder comparisons are appreciated and demonstrate that the learned representations contain structure beyond what linear probing captures. However, this raises a concern about fairness. The Kalman decoder introduces additional dynamic modeling parameters estimated on the finetune set, which goes beyond evaluating frozen representation quality via simple probing. Were the same non-linear/dynamic decoders also applied to the baseline methods (e.g., SBIND, NDT2, CEBRA)? Without this control, it is unclear whether the ZS improvement (0.18→0.29) reflects superior representation quality of WiCAT specifically, or whether similar gains would be observed with other representations when paired with a better decoder. Furthermore, I note an interesting pattern: Full FT+Linear (unfrozen backbone) achieves the highest finetune-set R² (0.58/0.42) but its ZS R² (0.37/0.24) is notably lower than KalmanSmooth with frozen backbone (0.43/0.29). This suggests that full finetuning may cause the representations to overfit to finetune subjects, while dynamic decoders on frozen representations better preserve cross-subject generalization. This observation deserves discussion, as it has practical implications for how the model should be deployed.
> >
> > 2. **[W1, Q1]**: The contextualization of R² relative to supervised within-session methods (R²~0.2–0.5) provides useful context. However, my original question (Q1) specifically asked about the theoretically achievable R² given the signal-to-noise ratio and neural variability inherent in widefield calcium imaging, essentially a noise ceiling analysis. The rebuttal instead compares against empirical R² values from existing methods, which reflects the limitations of those methods rather than the inherent information content of the data. A noise ceiling estimate (e.g., based on trial-to-trial reliability or split-half consistency of the neural signal) would more directly address whether 0.29 approaches the limits of what is decodable from widefield data, or whether substantial room for improvement remains. I would encourage the authors to include such an analysis or at minimum discuss why it may not be feasible.
> >
> > 3. **[W2]**: The authors acknowledge the data scale limitation and agree to add it to the Discussion. I appreciate the honest framing in the rebuttal ("first step toward foundation-style models"). However, the manuscript's abstract uses "Towards foundation modeling" and the introduction references "foundation-style modeling" in several places. With 2 datasets, 38 mice, and exclusively head-fixed paradigms, I would suggest the authors ensure that this cautious framing ("toward", "first step") is applied consistently throughout the manuscript. It is also worth considering whether the term "foundation" is appropriate at the current scale even as an aspirational qualifier, given the expectations this term carries in the broader ML community.
> >
> > 4. **[Q3]**: The explanation for AUD regions (low variance, proximity to atlas boundaries) is plausible but remains speculative without quantitative support. Reporting per-region signal variance statistics would strengthen this argument in the revision.
> >
> > 5. **[Q4]**: Table 11 only reports results on the Musall dataset. Given that the Kondo dataset has more subjects and greater cross-subject variability, verifying registration robustness on Kondo would be informative. Additionally, the text mentions scaling (0.9–1.1) and shift (±5 pixels) perturbations, but only rotation results appear in Table 11. Including the full perturbation results would provide a more complete picture.
> >
> > I maintain my current score. The paper makes a solid contribution to multi-subject widefield imaging modeling with a well-designed framework. However, the concerns above, particularly the lack of decoder-fairness controls in the new comparisons, the absence of a noise ceiling analysis for contextualizing ZS performance, and the gap between current experimental scale and the foundation-model terminology, prevent me from raising my assessment at this time.

---

> > > ### Author Response · Authors · 2026-04-06
> > >
> > > We thank the reviewer for their additional constructive comments and helpful insights. We address all points below.
> > >
> > > **Additional Attachment:** https://anonymous.4open.science/r/WiCAT/xFig.pdf
> > >
> > > ---
> > > # C1
> > > We now evaluate Kalman Smoother (KS) and MLP decoders for NDT2 and CEBRA, and a KS decoder for SBIND multi-session variants as nonlinear decoders were already used for SBIND in Table 1. Importantly, all models are probed with the same decoder in both Table 7 (LP) and **Table 12** (nonlinear/dynamic decoders) for fair comparison. While all models gain from nonlinear/dynamic decoders, WiCAT again maintains superior performance, particularly in the ZS setting, indicating stronger learned representations across decoders.
> > >
> > > We thank the reviewer for noting the interesting pattern. We agree that full finetuning can overemphasize subject-specific patterns, reducing transfer across subjects. Importantly, as the reviewer notes, dynamic decoders (KS) applied to frozen representations mitigate this effect, preserving generalization while improving ZS performance. We will add this interesting point to the discussion.
> > >
> > > **Table 12. Downstream Decoder Comparison Across Models (mean ± SEM).**
> > >
> > > Model|Musall FS|Musall ZS|Kondo FS|Kondo ZS
> > > -|-|-|-|-
> > > WiCAT+MLP|0.54±0.02|0.41±0.02|0.36±0.01|0.29±0.01
> > > WiCAT+KS|0.51±0.01|0.43±0.01|0.34±0.00|0.29±0.01
> > > NDT2+MLP|0.50±0.02|0.16±0.02|0.25±0.01|0.10±0.01
> > > NDT2+KS|0.49±0.02|0.16±0.01|0.25±0.01|0.10±0.01
> > > LocaNMF+CEBRA+MLP|0.48±0.01|0.25±0.04|0.29±0.01|0.03±0.01
> > > LocaNMF+CEBRA+KS|0.46±0.01|0.25±0.03|0.26±0.01|0.09±0.01
> > > SBINDUnsup+KS|0.46±0.01|0.28±0.04|0.28±0.01|0.17±0.01
> > > SBINDSup+KS|0.43±0.01|0.28±0.03|0.29±0.01|0.16±0.01
> > >
> > > ---
> > > # C2
> > > We thank the reviewer for this suggestion. We now include a noise ceiling analysis based on a split-half approach as suggested. For each session, trials are split into two halves, neural activity is averaged to obtain PSTHs, and their correlation provides an empirical upper bound on explainable neural variance, $R^2_{cap,neu}$. This yields ~0.95/0.34 (Musall/Kondo) (**xFig1**), reflecting strong trial alignment in Musall and weaker alignment in Kondo.
> > >
> > > While this provides an upper bound on neural reliability, translating this into a bound on behavior decoding R² requires separating the behavior-relevant and irrelevant components of neural signal, which in turn requires assumptions; we therefore report an approximate behavior-relevant bound under a linear projection of behavior onto neural activity. This step assumes linear neural-behavior relations and independence between signal, noise, and behavior-irrelevant activity (see attachment); while commonly used, these may not hold in practice, so we treat this bound as an informative approximation rather than a precise ceiling. We obtain $R^2_{cap,beh}$≈0.53/0.28 (Musall/Kondo).
> > >
> > > Across datasets, empirical linear regression (LR) performance remains below the bounds, consistent with their interpretation as upper limits for linear neural-behavior relations ($R^2_{LR}<R^2_{cap,beh}<R^2_{cap,neu}$) (xFig1). Overall, these analyses indicate that decoding limits are largely driven by data quality, and WiCAT achieves performance approaching $R^2_{cap,beh}$ in some settings while also consistently outperforming all baselines, indicating stronger learned representations.
> > >
> > > ---
> > > # C3
> > > We thank the reviewer for clarification regarding the use of “foundation” terminology.
> > > In the previous version, there were **6 instances** of “foundation” terminology, primarily referring to prior work or positioning our work as a step toward this direction, rather than claiming WiCAT itself as a foundation model. For clarity, we list all instances:
> > >
> > > L19: “...foundation-style modeling has been explored for some neural modalities… zero-shot…elusive in neurofoundation modeling...”
> > >
> > > L25: “Towards foundation modeling of widefield data, we introduce WiCAT…”
> > >
> > > L48: “One approach…is to develop multi-subject modeling…, thus constituting a step toward foundation-style modeling.”
> > >
> > > L57: “...foundation-style modeling…for other modalities…”
> > >
> > > L88: “We further demonstrate zero-shot…that has remained elusive in neurofoundation modeling across modalities.”
> > >
> > > To avoid ambiguity, we now **remove all of these occurrences**. We now refer to this concept only once in the Discussion, stating that this work represents a **first step toward foundation-style models** for widefield, consistent with the reviewer’s suggestion.
> > >
> > > ---
> > > # C4-5
> > > We now report region-wise variance and per-subregion statistics in xTable1. xFig2 further shows that boundary areas, including AUD, have very low VAR.
> > > ROI|VAR
> > > -|-
> > > SS|2.46
> > > VIS|1.15
> > > MO|1.25
> > > RSP|1.16
> > > OB|0.06
> > > AUD|0.04
> > >
> > > We clarify that in Table 11, all affine perturbations—scaling (0.9–1.1) and pixel shifts (±5 pixels)—are already applied jointly with the rotations listed for each row.  We now also report results on Kondo for completeness.
> > >
> > > **Table 12. Perturbation Effect on Kondo decoding R²**
> > > Rot|FS|ZS
> > > -|-|-
> > > 0°|0.34|0.18
> > > 10°|0.30|0.14
> > > 30°|0.29|0.13

---

### Official Review · Reviewer_9vGp · 2026-03-13

**Soundness:** 2
**Presentation:** 3
**Significance:** 3
**Originality:** 2
**Overall Recommendation:** 4
**Confidence:** 4

**Summary:**

This paper introduces WiCAT, a multi-subject self-supervised pretraining model for widefield calcium imaging. It leverages the Allen Brain Atlas to register recordings from different subjects into a unified spatial coordinate system, constructs a session-parameter-free shared representation space via spatiotemporal patch tokenization and global spatial embeddings, and pretrains with masked autoencoding (90% masking ratio). The model is evaluated on two public datasets (Musall: 13 mice / 26 sessions; Kondo: 25 mice / 352 sessions), demonstrating cross-subject behavior decoding, zero-shot behavior decoding, cross-dataset transfer, and zero-shot brain region reconstruction.

**Compliance With Llm Reviewing Policy:**

Affirmed.

**Final Justification:**

In the rebuttal, the authors have well addressed the prior concerns regarding the zero-shot performance and data diversity, etc. Currently, I lean to the positive point.

**Key Questions For Authors:**

1. Do the learned global spatial embeddings exhibit structure corresponding to known functional networks? I would find it very helpful if the authors could provide visualizations of the representations or attention patterns.
2. In what application scenarios is a zero-shot R² of 0.18–0.33 practically meaningful? Is there prior work that can serve as a reference for the practical significance of this performance level?
3. Why was no comparison made against LocaNMF + downstream model, the standard pipeline in the widefield imaging community?
4. How do the authors view data diversity vs. volume as the current scaling bottleneck? What are the expectations for further scaling?

**Limitations:**

1. Both datasets involve relatively simple sensorimotor tasks. I think it remains unclear whether the model would generalize to more complex cognitive tasks.

**Strengths And Weaknesses:**

Strengths：

1. This work addresses a clear and well-defined gap. Widefield calcium imaging currently lacks a multi-subject modeling framework. To my knowledge, this is the first foundation-style modeling attempt for this modality.
2. The core design is well-motivated. I find the use of the Allen CCF to eliminate inter-subject spatial variability without session-specific parameters to be biologically grounded and well-suited to widefield imaging, where cortical anatomy is highly consistent across individuals.
3. The experimental design is rigorous. The pretrain/finetune/zero-shot split ensures fair evaluation; the ablations cover the main design choices; and statistical significance is assessed via the Wilcoxon signed-rank test. I appreciate the thoroughness of this setup.

Weaknesses：

1. This paper lacks a scientific or application-level value proposition.** I think this is the most significant concern. The zero-shot R² reaches only 0.18–0.33, yet the paper does not discuss the practical significance of this performance level. More critically, the authors provide no scientific analysis of the learned representations (e.g., whether the spatial embeddings reflect known functional connectivity).
2. The data scale does not match the "foundation model" narrative. I find the current scale, 38 subjects and 378 sessions, insufficient to support a foundation model framing. The tests of cross-condition generalization are limited. Furthermore, Figure 5 shows finetune performance saturating after 30–50% of the data, which suggests to me that data diversity, rather than volume, may already be the bottleneck.
3. The baseline comparisons are insufficient. I note that the paper omits comparison with the standard widefield analysis pipeline (LocaNMF + downstream decoder). PCA + Linear Regression serves as an overly simplified baseline, and the multi-session extension of SBIND is author-implemented, which raises concerns about fairness in my assessment.
4. Minor issue. The paper repeatedly emphasizes that "zero-shot behavior decoding has remained elusive," but I would point out that widefield imaging inherently provides spatially consistent coverage across individuals. After atlas alignment, achieving zero-shot transfer is structurally far easier than in electrode-based recordings. I believe this advantage should be explicitly acknowledged to accurately scope the contribution.

---

> ### Author Rebuttal · Authors · 2026-03-31
>
> We thank the reviewer for their constructive feedback and for recognizing the well-defined gap addressed here and the biologically grounded design of our approach. We add new analyses to address their points.
>
> ---
> ## [W1, Q2] Zero-Shot (ZS) Performance
> We thank the reviewer for raising this important point. First, we include additional results (**new Table 10**; please see response to Reviewer YxAF, W3), where ZS performance improves to **R²≈0.29/0.43 (Kondo/Musall)** by including dynamics in our linear decoder (Kalman filtering).
> Second, absolute R² values should be interpreted in the context of widefield imaging. Due to strong spatiotemporal smoothing, noise, and substantial task-irrelevant activity, even fully supervised within-session decoding for widefield data typically reports R² values of ~0.2–0.5 (Musall et al.,2019; Saxena et al.,2020; Benisty et al.,2023; Hosseini et al.,2025). These works operate in an easier setting with access to subject-specific statistics, whereas our results are obtained on entirely unseen subjects.
>
> Thus, without access to subject-specific data, our fine-tune (FT) results surpass, and ZS results approach the range of supervised methods, suggesting meaningful predictive utility in practice, for example, for scientific studies such as comparing neural representations across animals. We next provide a demonstration of scientific interpretability through analyzing learned representations.
>
> ---
> ## [W1, Q1] Learned Representations Interpretation
> We agree that exploring the learned representations is important and add analyses in **new Figs. 17-18** (please see here: https://anonymous.4open.science/r/WiCAT/Figs.pdf).
> In Fig. 17, we apply t-SNE to patch embeddings, where each embedding is obtained by averaging over time and trials for each (session, patch). We observe that:
> * Across subjects, square patches from the same atlas-aligned grid location cluster together,
> * Neighboring patches are closer in latent space,
> * Clear left-right hemispheric symmetry (corresponding regions placed close together)
>
> We further show that patch embeddings are correlated according to anatomical structure (Fig. 18), with higher correlation for neighboring and homologous regions across hemispheres. These results indicate learned representations are **subject-invariant** and capture meaningful **anatomical organization**.
>
> ---
> ## [W2, Q4] Data Scale & Diversity
> We acknowledge that current publicly available widefield data does not yet match the scale of models in other domains. Our goal here is to develop a multi-subject framework with shared parameters for widefield imaging, where such approaches have not previously been demonstrated. This work thus provides a first step toward developing foundation-style models for widefield imaging as larger and more diverse datasets become available. We will add this point to the limitations.
> Here, we validate our cross-subject modeling framework on two of the largest publicly available datasets, which capture significant cross-subject and cross-session variability. We agree that the saturation trends reflect the current availability of large-scale public widefield data and expect that increasing task diversity as future datasets become available will shift this saturation point and allow scaling with both more data and higher model capacity. In our experiments, we observe that this saturation is not due to limited model capacity, suggesting that scaling with more diverse data is a promising direction.
>
> ---
> ## [W3] Baseline Comparisons
> We add comparisons with LocaNMF-based baselines in new Table 7. We apply LocaNMF on the pretraining set and apply the learned decomposition to FT and ZS sets. We then train decoders (CEBRA and linear regression (LR)) on FT set and evaluate on the test splits. WiCAT outperforms baselines, indicating that our self-supervised pretraining enables learning representations that generalize across subjects. For multi-session SBIND, we clarify that we use the original implementation without model changes; we simply align subjects to the atlas and train the same SBIND model across sessions.
>
> **Table 7. Decoding R² (mean±SEM) for Baselines**
> Model|Musall FS|Musall ZS|Kondo FS|Kondo ZS
> -|-|-|-|-
> WiCAT(Linear Probing)|0.51±0.01|0.33±0.01|0.34±0.00|0.18±0.01
> NDT2(Linear Probing)|0.48±0.02|0.13±0.01|0.25±0.01|0.07±0.01
> LocaNMF+CEBRA|0.46±0.01|0.24±0.03|0.24±0.01|0.09±0.01
> LocaNMF+LR|0.36±0.01|0.20±0.03|0.23±0.00|0.09±0.01
>
> ---
> ## [W4] Role of Atlas Alignment
> We agree that atlas alignment facilitates cross-subject transfer and makes this setting more structured compared to electrode-based recordings. We will add this to the Discussion.
> However, widefield imaging presents distinct challenges, including high dimensionality, task-irrelevant activity, and cross-subject variability from morphological differences and recording conditions. Our work addresses these challenges and shows that robust ZS continuous behavior decoding is achievable despite them.

---

> > ### Author Rebuttal · Reviewer_9vGp · 2026-04-04
> >
> > Thanks for the detailed feedback. Most of my concerns have been well addressed, I will update my rating accordingly.

---

> > > ### Author Response · Authors · 2026-04-08
> > >
> > > We thank the reviewer for their comments throughout the review process and for considering our response. We appreciate their positive feedback and their indication that they will update their score.

---

### Official Review · Reviewer_W6oc · 2026-03-13

**Soundness:** 2
**Presentation:** 4
**Significance:** 3
**Originality:** 3
**Overall Recommendation:** 4
**Confidence:** 4

**Summary:**

This paper is pretty timely and well position on the gorwing literature of foundation models in neural recordings and I'm finding that of interest to many readers in the field of neuroscience as well as applied AI community.

Although there has been cross-subject foundaiton models for other neural modalities in neurosceince (including eeg, fmri, and electrophysiology with targeted cortical region) can a bit weaken the weaken the position of the paper, as authors correctly claim, this practice has not been performed/published on widefield imaging modality.

The presentation and benchmarking of the paper is well done and enough technical details have been provided througought the paper and supplementaries.

**Compliance With Llm Reviewing Policy:**

Affirmed.

**Key Questions For Authors:**

1) can authors include some example plots showing the spatiotemrpal patterns from actual data and predicted motif ? if the claim of the paper is spatio-temporal prediction of wide field imagine then it would be natural to properly provide examples of spatio tmeporal map snapshot of the recording and comparison with the prediction to support the core claim. ( this is a common plot for widlefield imagine papers as authors might know!)

2) Are the codes for implementation provided ? Since the validation of this claim highly depends on successful code and implementation (in contrast to papers with theoretical novel ideas, or conceptual novel frameworks) the importance of open access code (or even parital implementation) is higher for this type of works.

3) What are the chance levels for the prediction decoding scores in figure 3? in general providing the chance levels would be helpful to asses the degree of complexity of the decoding task.

4) as authors know the autocoronations of widefield imagine is high both in space and time. Specially regarding the spatial prediction of unobserved regions, there is a concern that high autocoronations can enhance the chance level value for decoding.

**Limitations:**

Lack of open access code or repository for the trained networks keep the paper in a safe position of not being able to be replocated nor used by larger audience.
The "modality specific challenges" in decoding could be discussed in comparison to existing similar foundation models for more common modalities.

**Strengths And Weaknesses:**

Strength:
proper technical details, clear writing and presenation, solid benchmarking and comparison across methods. Basically regardles of the singificnace of the question, the paper has all the required component for a proper computational neurosceince method paper for AI/ML venues.


Weakness:

The significance of the method could enhance if the paper was comparing this cross-subject foundation model in tehcnical aspect with other similar work on different neural modalities.

The effective temporal horizon for prediction could be better explored by additional simulations and the spatio-temporal maps could be attahced at least in the appendix beyound time traces of activities across selected unobserved regions.

---

> ### Author Rebuttal · Authors · 2026-03-31
>
> We thank the reviewer for their constructive feedback and for recognizing the relevance and timeliness of our work. We incorporate additional experiments to address their points.
>
> ---
> ## [W1] Baseline Comparisons
>
> We add a comparison to NDT2 (Ye et al., 2023), a recent cross-subject model of electrophysiology (new Table 7; please see Reviewer 9vGp, W3).
> To enable fair comparison, we adapt NDT2 (designed for spiking data) to imaging data by:
> * Replacing the spike embedding layer with a convolutional tokenizer mapping each $32\times32$ patch (patch size=1024) to a token.
> * Replacing original Poisson likelihood with MSE loss for imaging data.
>
> Since NDT2 relies on session- and subject-specific embeddings, we include finetune (FT) sessions during NDT2 pretraining to learn the subject/session ids for them. Similar to WiCAT’s linear probing, for NDT2 finetuning, we use a linear decoder to predict behavior for each dataset. For zero-shot (ZS) evaluation, we use subject/session ids from a random subject/session seen during pretraining. Otherwise, we use default NDT2 settings. Results (Table 7) show that WiCAT consistently outperforms NDT2, particularly in the ZS setting, highlighting the advantage of learning shared representations without session-specific parameters.
>
> ---
> ## [W2, Q1] Spatiotemporal Predictions
> We add R² maps for left-out region prediction (**new Fig. 14**), and reconstruction snapshots for ZS subjects (**new Fig. 15**) in the appendix (also shared via the link: https://anonymous.4open.science/r/WiCAT/Figs.pdf). These results show that:
> * WiCAT captures global spatial patterns during pretraining,
> * These patterns transfer effectively to ZS subjects, maintaining fidelity even when both regions and subjects are unseen.
>
> **Temporal horizon analysis:** We evaluated long-range prediction by:
> * Providing only the first ~110 frames of each trial as context,
> * Predicting neural activity for the remaining 100 frames.
> As shown in **new Fig. 16**, while performance degrades over longer horizons, WiCAT remains robust in ZS settings, with minimal additional drop when generalizing to unseen subjects.
>
> ---
> ## [Q2] Code Availability
>
> We agree that sharing the implementation is important and provide an **anonymous repository** containing the model and pretrained weights (https://anonymous.4open.science/r/WiCAT). All code will be publicly released upon acceptance.
>
> ---
> ## [Q3] Chance-Level Decoding
> We thank the reviewer for raising this important point to contextualize the reported results. We now include chance-level baselines computed via permutation tests. Specifically, we shuffle behavioral signals across trials (i.e., reassign entire trial-length behavior sequences to different trials) within each session (1000 permutations), while keeping the neural activity unchanged. This preserves trial-level correlations while breaking neural-behavioral alignment, yielding a strong/conservative chance-level baseline. We then fit a PCA+regression model using the same evaluation pipeline.
>
> **Table 8. Chance Levels Decoding (R², mean±SEM over sessions)**
> ||FT|ZS
> -|-|-
> Musall|0.23±0.02|0.11±0.04
> Kondo|0.00±0.00|0.00±0.00
>
> These results show some correlation due to task structure in the Musall dataset. However, for Kondo, where animal behavior is less constrained, chance levels are essentially zero. This confirms that WiCAT’s performance is significantly above chance (one-sided permutation test, N=1000, p<0.001 for all sessions). We further note that in the rebuttals, we now include improved ZS analyses, which achieve substantially higher ZS decoding performance (see **new Table 10** in Reviewer YxAF, W3).
>
> ---
> ## [Q4] Chance-Level Neural Prediction
> Similar to Q3, we compute chance-level baselines for the cross-region neural prediction task. Specifically, for each target ROI, we fix the source brain regions and shuffle trial-length blocks of the target ROI activity across trials (similar to the approach in Q3) within each session (10 permutations). We then fit the same linear decoder baseline used in the main experiments. WiCAT is significantly better than chance for all ROIs (**new Table 9**; one-sided Wilcoxon signed-rank test, n=168, p<0.001 for all regions).
>
> **Table 9. ZS Chance-Level Neural Prediction (MSE, mean±SEM over sessions)**
> ROI|MSE
> -|-
> SS|2.44±0.05
> VIS|1.14±0.00
> MO|1.09±0.03
> RSP|1.12±0.02
> OB|0.06±0.01
> AUD|0.06±0.00
>
> ---
> ## [L1] Modality-Specific Challenges
> We agree that it is important to contextualize widefield imaging relative to other modalities and will clarify these aspects in the manuscript. Compared to spiking data, widefield imaging is high-dimensional with substantial task-irrelevant activity across the cortex, spatially smooth and correlated, and has local and global cortex-wide spatiotemporal dependencies, leading to modality-specific challenges for decoding and representation learning. In addition, variability in alignment and morphology across subjects introduces noise in cross-subject settings.

---

> > ### Author Rebuttal · Reviewer_W6oc · 2026-04-04
> >
> > Appreciate the authors response and incorporating the 1) code, 2) spatial pattern prediction which is important for WF imagine and proper interpretation of them, 3) and clarification of Chance level prediction, although it would be meaningful to have them updated in the corresponding figures as well for the camera ready.
> >
> > I update my confidence score in my assessment of the positive overall score provided.

---

> > > ### Author Response · Authors · 2026-04-08
> > >
> > > We thank the reviewer for their thoughtful comments throughout the review process and for considering our response. We will definitely add the provided analyses for the reviewer to the revised manuscript. We appreciate that they are updating their confidence score in their overall positive view of our work.

---

### Decision · Program_Chairs · 2026-04-30

**Decision:**

Accept (regular)

**Comment:**

The paper introduces a self-supervised pretraining model for widefield calcium imaging. The model constructs shared representation space via spatiotemporal patch tokenization and global spatial embeddings, and pretrains with masked autoencoding. The model is evaluated on two public datasets. The reviewers positeively commented that the paper addresses a clear and well-defined gap, as widefield calcium imaging seems to lack a multi-subject modeling framework. The use of the Allen CCF to eliminate inter-subject spatial variability without session-specific parameters was seen to be biologically grounded and well-suited to widefield imaging, where cortical anatomy is highly consistent across individuals. Also, the experimental design was seen as rigorous. Critical comments were made regarding the scale of the pretraining dataset, the lack of comparison with other neural foundation models, as well as low zero shot performance in some dataset. While the rebuttal phase was able to clarify many of the questions, the reviews convey a mixed view, with three weak accepts and one weak reject. In light of the overall positive reviews, the AC suggests **accept**.